# Discrimination-Free Insurance Pricing with Privatized Sensitive Attributes

## Abstract

Fairness has emerged as a critical consideration in the landscape of machine learning algorithms, particularly as AI continues to transform decision-making across societal domains. To ensure that these algorithms are free from bias and do not discriminate against individuals based on sensitive attributes such as gender and race, the field of algorithmic bias has introduced various fairness concepts, along with methodologies to achieve these notions in different contexts. Despite the rapid advancement, not all sectors have embraced these fairness principles to the same extent. One specific sector that merits attention in this regard is insurance. Within the realm of insurance pricing, fairness is defined through a distinct and specialized framework. Consequently, achieving fairness according to established notions does not automatically ensure fair pricing in insurance. In particular, regulators are increasingly emphasizing transparency in pricing algorithms and imposing constraints on insurance companies on the collection and utilization of sensitive consumer attributes. These factors present additional challenges in the implementation of fairness in pricing algorithms. To address these complexities and comply with regulatory demands, we propose an efficient method for constructing fair models that are tailored to the insurance domain, using only privatized sensitive attributes. Notably, our approach ensures statistical guarantees, does not require direct access to sensitive attributes, and adapts to varying transparency requirements, addressing regulatory demands while ensuring fairness in insurance pricing.

## 1 Introduction

Fairness has emerged as a critical consideration in the landscape of machine learning algorithms. Various concepts of algorithmic fairness have been established in this burgeoning field including demographic parity, equalized odds, predictive parity, among others (Calders et al., 2009; Dwork et al., 2012; Feldman, 2015; Hardt et al., 2016; Zafar et al., 2017; Kusner et al., 2018). It is essential to emphasize that not all of these metrics are universally applicable to every situation. Each fairness concept bears its own merits that align with specific contextual applications (Barocas et al., 2019). In addition to the theoretical underpinnings of fairness notations, the literature has also witnessed a substantial development of methodologies in achieving various fairness criteria.

In contrast to algorithmic fairness, the insurance industry employs a unique and specialized framework, known as actuarial fairness. This well-established concept serves as a fundamental principle in pricing insurance contracts (Frees & Huang, 2023). The premium is considered actuarially fair if it is a sound estimate of the expected value of all future costs associated with an individual risk transfer (CAS, 2021). Given the stringent regulatory environment, insurers are mandated to demonstrate actuarial fairness in their premiums. As machine learning algorithms become more prevalent in insurance company operations, regulatory bodies in recent years have begun to reassess the concept of fairness, in particular, questioning whether an actuarially fair premium should discriminate against policyholders based on sensitive attributes, such as gender and ethnicity. For instance, Directive 2004/113/EC ("Gender Directive") issued by the Council of the European Union prohibits insurance companies in the UE from using gender as a rating factor for pricing insurance products (Xin & Huang, 2023). More recently, the governor of the state of Colorado signed Senate Bill (SB) 21-169 into law, protecting consumers from insurance practices with unfair discrimination on the basis of race, color, national or ethnic origin, religion, sex, sexual orientation, disability, gender identify, or gender expression. Under this backdrop, our research aims to develop a method enabling insurers to integrate machine learning algorithms in the context of insurance pricing while adhering to the regulatory mandates regarding fairness, transparency, and privacy. As underscored by Lindholm et al.

(2022b) the actuarial fairness and algorithmic fairness may not coexist simultaneously under certain conditions. Consequently, our focus is on the discrimination-free premium, a conceptual framework recently introduced in the actuarial science literature. This discrimination-free premium, aligned with the notion of fairness from a causal inference perspective, is free from both direct and indirect discrimination linked to sensitive attributes (Lindholm et al., 2022a).

We consider a multi-party training framework, where the insurer has direct access to non-sensitive attributes of policyholders but lacks access to the true sensitive attributes. Instead, a noised or privatized version of sensitive attributes is securely stored with a trusted third party (TTP). The central premise of our method is that the insurer forwards transformed non-sensitive attributes and the response variable to the TTP. Then, TTP combines the privatized sensitive attributes and information provided by the insurer to train a machine learning model. The resulting discrimination-free premium is then transmitted back to the insurer (See Figure 1). The multi-party framework is driven by two key practical considerations: First, because of the regulatory constraints, insurance companies are either prohibited from directly accessing sensitive attributes or are limited to accessing only a noised version of such attributes. Second, as sophisticated AI techniques become more prevalent, insurers are increasingly turning to third-party vendors to implement complex machine learning methods.

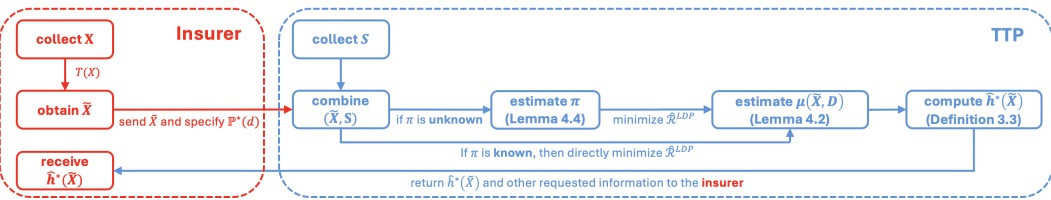

Figure 1: Insurer-TTP Interaction Diagram

In our method, the noise in sensitive attributes can arise in various scenarios including but not limited to: 1) Data collection mechanisms: Privacy filters used by insurers or third parties to encourage data sharing introduce distortion to protect privacy. 2) Measurement errors: Errors in sensitive attributes can originate from either policyholders or insurers. Policyholders may provide inaccurate information, and insurers may impute missing values, both leading to errors. 3) Privatization for data transmission security: Sensitive attributes are privatized for secure transmission between insurers and third parties.

Our multi-party training framework is general and includes two scenarios as special cases: First, the insurer obtains privatized sensitive attributes from a third party and applies the algorithm directly; Second, the insurer collects data on both sensitive and non-sensitive attributes and outsources the pricing algorithm to a third party. We emphasize that our proposed framework is both theoretically sound and practical. It aligns seamlessly with the well-established insurer-TTP protocols already in place in the insurance market, enabling straightforward implementation. For instance, major insurers with in-house pricing capabilities often supplement their proprietary data with third-party data. In such cases, insurers can leverage existing protocols to obtain sensitive attributes from third parties. In contrast, small to mid-sized insurers commonly rely on industry-wide data, process it using credibility techniques, and then transfer the processed data to a data service platform for pricing. In this context, the proposed method can be applied by enabling the insurer to collect both sensitive and non-sensitive data and forward it to a third-party vendor for pricing algorithm execution.

In our study, we consider two practical scenarios: 1) Known noise rate: TTP has full information regarding the privatized sensitive attributes, including both the privacy mechanism and the noise rate. 2) Unknown noise rate: TTP has access to the privatized sensitive attributes, with knowledge limited to the privacy mechanism and no information about the noise rate. The proposed method enjoys several advantages: 1) The insurer does not need direct access to sensitive attributes to implement the method. 2) The method solely relies on the privatized sensitive attributes, irrespective of the entity responsible for gathering such information. 3) The method is straightforward to implement and provides statistical assurance. In the pursuit of the actuarial fairness proposed by Lindholm et al. (2022a), our contributions are threefold: 1) We introduce an efficient method to train discrimination-free models that are transparency-adaptive. Notably, it only requires access to privatized sensitive attributes. 2) We provide statistical assurances both when the noise rate for the privacy mechanism is known and unknown. 3) We demonstrate the empirical effectiveness of our method and provide insight into the effect of noise rate estimation error on our proposed method.

## 2 BACKGROUND AND RELATED WORK

### 2.1 FAIRNESS IN MACHINE LEARNING

Algorithmic fairness literature primarily distinguishes between two types: individual fairness (Dwork et al., 2012; Barocas et al., 2019) and group fairness (Kamishima et al., 2012; Feldman, 2015; Friedler et al., 2018). Individual fairness emphasizes the idea that similar individuals should be treated similarly. It focuses on ensuring that the predictions or outcomes of the algorithm are consistent for individuals who share similar characteristics, regardless of their belonging to any specific group. Group fairness, by contrast, seeks equitable treatment across predefined demographic groups, such as race or gender. While individual fairness may imply group fairness under conditions (Dwork et al., 2012), they are often studied separately. The discrimination-free premium framework aligns more closely with the principles of individual fairness, though it does not fall strictly within either category.

Fair model training methods are generally classified into three categories: pre-processing, where fairness is enforced on the training data before using it to train machine learning models (Adebayo & Kagal, 2016; Calmon et al., 2017; Plečko & Meinshausen, 2019); in-processing, which incorporates fairness constraints during training (Agarwal et al., 2018; 2019; Donini et al., 2020); and post-processing, which enforces fairness during inference on an already trained model (Hardt et al., 2016; Woodworth et al., 2017). Our proposed method shares similarities with a post-processing approach, albeit with subtle yet significant differences. Specifically, post-process methods typically formulate the fairness problem as a constraint optimization. However, achieving the fairness notation proposed by Lindholm et al. (2022a) in insurance pricing is incompatible with this framework. As a result, it is crucial to recognize that techniques commonly employed in post-processing are not readily applicable in the insurance pricing setting. Our work utilizes group-specific loss that shares a similar idea to the decoupling classifier studied by Dwork et al. (2018); Ustun et al. (2019), yet is formulated very differently. Furthermore, our work has connections to learning under corrupted features, as explored in studies like Li et al. (2016); van der Maaten et al. (2013). In contrast to these studies, our method offers two key advantages: 1) its versatility, as it is compatible with any valid loss function based on our noise setup, and 2) its simplicity, as it is very easy to implement, setting it apart from previous approaches in this domain.

### 2.2 FAIRNESS IN INSURANCE PRICING

With the advent of deep learning in assisting insurance pricing, actuarially fair premiums can be more accurately estimated (Shi et al. (2024)). However, regulators have raised concerns about potential discrimination based on sensitive attributes like gender and ethnicity, prompting a reevaluation of fairness in actuarial science. Conceptually, studies such as Shimao & Huang (2022); Xin & Huang (2023); Frees & Huang (2023) have explored fairness in insurance, distinguishing between two types of discrimination: direct discrimination, where sensitive attributes are explicitly used as rating factors, and indirect discrimination (or proxy discrimination), where non-sensitive attributes serve as proxies for sensitive ones. Methodologically, fair pricing models have been developed using three main approaches: 1) the counterfactual method rooted in causal statistics Iturria et al. (2022). 2) group fairness methods akin to those in algorithmic fairness Grari et al. (2022), and 3) the probabilistic approach Lindholm et al. (2023). However, these approaches rely on direct access to true sensitive attributes, a practice that may not align with the progressively stringent regulatory environment in insurance. In contrast, our work addresses this limitation by using only noisy versions of sensitive attributes, offering a novel framework for training discrimination-free insurance pricing models that comply with regulatory constraints. To the best of our knowledge, this is among the first attempts to tackle these real-world challenges.

## 3 PRELIMINARIES & PROBLEM FORMULATION

Consider $n$ i.i.d triplets $\{X_i, D_i, Y_i\}_{i=1}^n$ drawn from an unknown distribution $(X_i, D_i, Y_i) \sim \mathcal{P}$, where $X_i \in \mathcal{X}$ are the non-sensitive attributes, $D_i \in \mathcal{D}$ are the true sensitive attributes which we consider to be discrete, and $Y_i \in \mathcal{Y}$ are the outcome of interest which can be either continuous or discrete. In the rest of the paper, we use the below definitions on insurance price:

**Definition 3.1.** Best-estimated Price: the best-estimated price for $Y$ w.r.t. $(X, D)$ is defined as:

$$\mu(X, D) := \mathbb{E}[Y|X, D].$$

This price directly discriminates policyholders based on their sensitive attributes, because $D$ is explicitly used in the calculation of insurance premiums.

**Definition 3.2.** Unawareness Price: the unawareness price for $Y$ w.r.t. $X$ is defined as:

$$\mu(X) := \mathbb{E}[Y|X].$$

Although $\mu(X)$ does not explicitly depend on $D$, it is a price with indirect discrimination because one can potentially infer $D$ from $X$ when they are correlated. To see this,

$$\mu(X) = \int_d \mu(X, d) d\mathbb{P}(d|X).$$

**Definition 3.3.** Discrimination-free Price: the discrimination-free price for $Y$ w.r.t. $X$ is defined as:

$$h^*(X) := \int_d \mu(X, d) d\mathbb{P}^*(d),$$

where $\mathbb{P}^*(d)$ is defined on the same range as the marginal distribution of $D$.

Note that if $Y \perp D|X$ holds, the unawareness price coincides with the discrimination-free price. However, this condition often fails in practice due to omitted variables. In such cases, it is essential to reorganize the trade-off between the two types of fairness embedded in the discrimination-free premium $h^*(X)$, the equity-based and the risk-based fairness. The former aims to prevent differentiation in pricing based on the sensitive attribute $D$, while the latter concerns the across-group subsidization that potentially arises when the sensitive attribute $D$ is excluded from pricing. The overall value of the discrimination-free premium ultimately depends on the extent of cross-subsidization. In practice, insurers address this issue by gathering additional data from various sources to account for potential omitted variables in their pricing models.

## 4 DISCRIMINATION-FREE PRICING FOR INSURANCE

The ultimate goal is to compute the discrimination-free premium $h^*(X)$ which is further determined by two components, namely $\mu(X, D)$ and $\mathbb{P}^*(d)$. In our framework, a straightforward choice for $\mathbb{P}^*(d)$ is its empirical distribution. More generally, we can view it as a turning parameter to satisfy some desired statistical criteria (e.g. unbiasedness). Therefore our primary concern lies on $\mu(X, D)$, and in the following sections, we discuss its estimation under both true and noised sensitive attributes.

### 4.1 PRICING UNDER TRUE SENSITIVE ATTRIBUTES

There are two parties, namely the insurer and a trusted third party (TTP). In the first step, the insurer applies some transformation $T$ on $X$, denoted as $\tilde{X} := T(X)$. Then the insurer passes the transformed data $\{\tilde{X}, Y\}$ to the TTP. In the second step, the TTP estimates $\mu(\tilde{X}, D)$ and computes $h^*(\tilde{X})$ following Definition 3.3. Then, TTP returns $\mu(\tilde{X}, D), h^*(\tilde{X})$ to the insurer.

Let $f_k \in \mathcal{F}, \forall k \in [|\mathcal{D}|]$, where $\mathcal{F}$ is a hypothesis class and $f_k : T(\mathcal{X}) \to \mathbb{R}_+, \forall k \in [|\mathcal{D}|]$ is a score function. The TTP learns $\mu(\tilde{X}, D)$ by minimizing the expected risk:

$$\mathcal{R}(f_1, \ldots, f_{|\mathcal{D}|}) = \sum_{k=1}^{|\mathcal{D}|} \left( \mathbb{E}_{Y, \tilde{X}|D=k} \left[ L(f_k(\tilde{X}), Y) \right] \cdot \mathbb{P}(D = k) \right), \tag{1}$$

for a generic loss function $L$ and we denote $\mathcal{R}(f_k) = \mathbb{E}_{Y, \tilde{X}|D=k}[L(f_k(\tilde{X}, Y))]$. The learning process is generally applicable, as there are no restrictions on the transformation $T$, hypothesis class $\mathcal{F}$, and the loss function $L$. Using a pre-specified $\mathbb{P}^*(d)$, the TTP then computes $h^*(\tilde{X})$ by

$$h^*(\tilde{X}) = \sum_{k=1}^{|\mathcal{D}|} f_k(\tilde{X}) \cdot \mathbb{P}^*(D = k). \tag{2}$$

The above procedure is summarized in an algorithmic manner (MPTP) in Appendix A.

**Remark 1:** The framework centers on the specification of group-specific score functions $f_1, \ldots, f_k, \forall k \in [|\mathcal{D}|]$, which provides two key advantages: 1) The framework naturally extends

when sensitive attributes are privatized. 2) The computation of $h^*(\tilde{X})$ does not require the disclosure of group membership information $D$, enabling its implementation by either the TTP or the insurer.

**Remark 2:** There is an intrinsic trade-off between model transparency and model complexity. In our framework, they are governed by both the insurer (via transformation $T$) and the TTP (via hypothesis class $\mathcal{F}$). For example, when $T$ is the identity transformation and $\mathcal{F}$ is the class of linear models, we achieve the maximum model transparency as it simplifies to a (generalized) linear model w.r.t. $X$.

**Remark 3:** The optimized $f_k$'s are independent of the specific form of $h^*$. More generally, they can be directly applied to any downstream tasks that do not depend on the optimization procedure of $f_k$'s.

**Example:** An insurer employs a GLM based on the exponential dispersion family to model insurance claims. The deviance loss is used by both the insurer and the TTP:

$$L = -2\phi(\ell(\mu, \phi) - \ell_s).$$

For $n$ i.i.d. triplets $\{X_i, Y_i, D_i\}_{i=1}^n$ drawn from an unknown population $\mathcal{P}$, the insurer only observes $\{X_i, Y_i\}_{i=1}^n$, and the TTP observes $\{D_i\}_{i=1}^n$. The insurer constructs $\tilde{X}_i = T(X_i)$ using a feed-forward neural network. Let $h \in \mathcal{H}$ where $\mathcal{H}$ is a hypothesis class and $h : \mathcal{X} \to \mathbb{R}_+$ is a score function. Suppose the neural network consists of $m$ layers, and there are $q_m$ hidden nodes in the $m^{\text{th}}$ layer. For $\mathcal{X} \in \mathbb{R}^{q_0}$, denote the composition $z^{(m:1)} : \mathbb{R}^{q_0} \to \mathbb{R}^{q_m}$, where $z^{(j)} : \mathbb{R}^{q_{j-1}} \to \mathbb{R}^{q_j}, \forall j \in [m]$. Then, the transformation is:

$$T(X_i) = z^{(m:1)}(X_i) = z^{(m)}(z^{(m-1)}(\cdots(z^{(1)}(X_i)))),$$

which is learned by the insurer via minimizing the empirical risk:

$$\hat{\mathcal{R}}(h) = \sum_{i=1}^n L(h(X_i), Y_i). \tag{3}$$

After obtaining $\{\tilde{X}\}_{i=1}^n$ (an $n \times q_m$ matrix), the insurer passes them to the TTP along with $\{Y_i\}_{i=1}^n$. The TTP first estimates $\mu(\tilde{X}_i, D = k) = f_k(\tilde{X}_i)$ by minimizing the empirical risk:

$$\hat{\mathcal{R}}(f_1, \ldots, f_k) = \sum_{i=1}^n \sum_{k=1}^{|\mathcal{D}|} L(f_k(\tilde{X}_i), Y_i) \cdot \mathbf{1}\{D_i = k\}, \tag{4}$$

As an example, $f_k$ could be specified as a linear model such that:

$$f_k(\tilde{X}_i)) = \beta_0^k + z(\tilde{X}_i)_1 \beta_1^k + \cdots + z(\tilde{X}_i)_{q_m} \beta_{q_m}^k.$$

Then the TTP calculates the discrimination-free price following Definition 3.3:

$$\hat{h}^*(\tilde{X}_i) = \sum_{k=1}^{|\mathcal{D}|} \hat{f}_k(\tilde{X}_i) \cdot \hat{\mathbb{P}}(D = k), \tag{5}$$

and returns $\{\hat{\mu}(\tilde{X}_i, D), \hat{h}^*(\tilde{X}_i)\}_{i=1}^n$ to the insurer.

**Remark:** In this example, the insurer obtains $T$ by training a neural network. Nonetheless, there are no constraints on how the insurer constructs $T$. In principle, $\tilde{X}_i$ could be engineered features that are learned through any supervised or unsupervised method, or $\tilde{X}_i$ could be privatized features designed specifically for the purpose of secure data transmission.

### 4.2 PRICING UNDER PRIVATIZED SENSITIVE ATTRIBUTES WITH KNOWN NOISE RATES

In this section, we investigate a scenario where the true sensitive attributes are not directly observable by the TTP. Instead, the TTP only has access to their privatized versions. This situation often occurs in practice when privacy-preserving mechanisms are employed during data collection or transmission stages (refer to Section 1 for detailed discussions). The central inquiry revolves around how the TTP can obtain a fair price, which entails minimization of Eq. (1), without direct access to $D$.

To address this challenge, we employ the concept of local differential privacy (LDP) in our framework. Let $S$ denote the privatized sensitive attributes. The $\epsilon$-LDP mechanism $Q$ is defined as:

**Definition 4.1.**

$$\max_{s,d,d'} \frac{Q(S = d|d)}{Q(S = s|d')} \le e^\epsilon.$$

Under the randomized response mechanism in Warner (1965); Kairouz et al. (2015), one has:

$$Q(s|d) = \begin{cases} \frac{e^\epsilon}{|\mathcal{D}|-1+e^\epsilon} := \pi, \text{if } s = d \\ \frac{1}{|\mathcal{D}|-1+e^\epsilon} := \bar{\pi}, \text{if } s \neq d, \end{cases}$$

where $|\mathcal{D}|$ is the cardinality of $\mathcal{D}$ and $s$ is sampled from $Q(\cdot|d)$ independently from $X, Y$.

The primary advantage of employing LDP is that the data collector cannot definitely ascertain the true value of sensitive attributes, irrespective of the accuracy of the information provided for any observation in the dataset (Mozannar et al., 2020). Consequently, any model trained on this dataset preserves differential privacy with respect to the sensitive attributes.

Similar to the setup in Section 4.1, the insurer observes $\{X_i, Y_i\}_{i=1}^n$, provides a transformation $T$, and passes $\{\tilde{X}_i, Y_i\}_{i=1}^n$ to the TTP. The TTP is to estimate $\mu(X, D)$ by combining the data from the insurer with privatized sensitive attributes $S_i$. Lemma 4.2 establishes a population equivalent risk under privacy mechanism $Q(s_i|d_i)$ and Theorem 4.3 provides the associated statistical guarantees.

**Lemma 4.2.** *Given the privacy parameter $\epsilon$, minimizing the risk (Risk-LDP) defined by Eq. (6) under $\epsilon$-LDP w.r.t. privatized sensitive attributes $S$ is equivalent to minimizing Eq. (1) w.r.t. true sensitive attributes $D$ at the population level:*

$$\mathcal{R}^{LDP}(f_1, \ldots, f_k) = \sum_{k=1}^{|\mathcal{D}|} \sum_{j=1}^{|\mathcal{D}|} \left( \mathbf{\Pi}_{kj}^{-1} \mathbb{E}_{Y, \tilde{X}|S=j} \left[ L(f_k(\tilde{X}), Y) \right] \cdot \sum_{l=1}^{|\mathcal{D}|} \mathbf{T}_{kl}^{-1} \mathbb{P}(S = l) \right), \quad (6)$$

*where $\mathbf{\Pi}^{-1}$ and $\mathbf{T}^{-1}$ are $|\mathcal{D}| \times |\mathcal{D}|$ row-stochastic matrics.*

Empirically, for a given policyholder $i$, the TTP computes $\hat{h}^*(\tilde{X}_i)$ using learned $\{\hat{f}_k(\tilde{X}_i)\}_{k=1}^{|\mathcal{D}|}$, and returns $\hat{h}^*(\tilde{X}_i)$ and $\hat{\mu}(\tilde{X}_i, D = k)$ for $k = 1, \ldots, |\mathcal{D}|$ to the insurer. We also summarize the above procedure (MPTP-LDP) in an algorithmic manner (see Appendix A).

**Remark:** The use of group-specific score functions enables straightforward construction of an equivalent risk for Eq. (1) using only $S_i$. It is crucial not to view it as a limitation of our approach. As discussed in Section 2, achieving a closed-form equivalence is not always feasible with a conventional score function $f(\tilde{X}, D)$. Existing methods tackling similar challenges often rely on surrogate risks or confine themselves to specific loss functions (Li et al., 2016; Al-Rubaie & Chang, 2019).

**Theorem 4.3.** *For any $\delta \in (0, \frac{1}{2})$, $C_1 = \frac{\pi + |\mathcal{D}| - 2}{|\mathcal{D}|\pi - 1}$, denote $VC(\mathcal{F})$ as the VC-dimension of the hypothesis class $\mathcal{F}$, and $K$ be some constant that depends on $VC(\mathcal{F})$. Let $f = \{f_k\}_{k=1}^{|\mathcal{D}|}$ where $f_k \in \mathcal{F}$ and let $L : Y \times Y \to \mathbb{R}_+$ be a loss function bounded by some constant $M$. Denote $k^* \leftarrow \arg\max_k |\hat{\mathcal{R}}^{LDP}(f_k)\hat{\mathbb{P}}(D = k) - \mathcal{R}^{LDP}(f_k)\mathbb{P}(D = k)|$. If $n \geq \frac{8 \ln(\frac{|\mathcal{D}|}{\delta})}{\min_k \mathbb{P}(S=k)}$, then with probability $1 - 2\delta$:*

$$\hat{\mathcal{R}}^{LDP}(f) \leq \mathcal{R}(f^*) + K\sqrt{\frac{VC(\mathcal{F}) + \ln(\frac{\delta}{2})}{2n}} \frac{2C_1 M |\mathcal{D}|}{\mathbb{P}(S = k^*)}.$$

**Remark:** The bound grows with $|\mathcal{D}|$. However, in insurance practice, $|\mathcal{D}|$ is small in most cases. When $|\mathcal{D}|$ is large, categorical embedding (see Shi & Shi (2023)) can be applied if regulation permits.

### 4.3 Pricing under Privatized Sensitive Attributes with Unknown Noise Rates

This section expands upon the discussion in Section 4.2 to address scenarios where the noise rate of the privacy mechanism is not known a priori. It is essential to note that constructing a population-equivalent risk requires knowledge of $\pi, \bar{\pi}$. However, obtaining such information often proves challenging in practice, particularly when the sensitive attributes are subject to measurement errors (refer to Section 1 for detailed discussions). Within our multi-party framework, we consider a setup akin to that of Section 4.2, with the key distinction being that the TTP lacks knowledge of the true conditional probabilities $\pi$ and $\bar{\pi}$ for the given privacy mechanism $Q(s_i|d_i)$. To tackle this obstacle, we propose a methodology wherein the TTP first estimates $\pi$ and $\bar{\pi}$ from the data and then uses these estimates to construct the population-equivalent risk, following the approach outlined in Section 4.2. We summarize the estimation procedure for $\pi$ and $\bar{\pi}$ in Lemma 4.4 and delineate the underlying assumptions that underpin the establishment of statistical guarantees in Theorem 4.5.

**Lemma 4.4.** *Consider $\epsilon$-LDP setting with $\pi \in (\frac{1}{|\mathcal{D}|}, 1]$ and $\bar{\pi} \in [0, \frac{1}{|\mathcal{D}|})$. For some transformation of $X$, denoted by $X^* = \tilde{T}(X)$, assume there exists at least one anchor point $X^*_{anchor}$ in the dataset s.t. $\mathbb{P}(D = j^*|X^*_{anchor}) = 1$ for some $j^* \in [|\mathcal{D}|]$. Then $\pi = \mathbb{P}(S = j^*|X^*_{anchor})$. Empirically, for a dataset with $n$ observation, let $\boldsymbol{\eta}^n_{j^*}(X^*) = \left( \hat{\mathbb{P}}(S = j^*|X^*_1), \ldots, \hat{\mathbb{P}}(S = j^*|X^*_n) \right)$, then $\hat{\pi} = \left\| \boldsymbol{\eta}^n_{j^*}(X^*) \right\|_\infty$.*

Besides Lemma 4.4, we introduce the assumptions and procedure to establish Theorem 4.5 in the following (A more detailed discussion is in Appendix B):

**Step 1: Grouping:** we evenly divide $\{X^*_i, S_i\}^n_{i=1}$, into $n_1$ groups, with $m = \frac{n}{n_1}$ samples each.
**Step 2: Estimating within groups:** for any $k \in [n_1]$, within each group $\{X^*_{k,j}, S_{k,j}\}^m_{j=1}$, we then derive an $m$-dimension vector $\boldsymbol{\eta}^m_{j^*,k}(X^*_{k,\cdot}) = \left( \hat{\mathbb{P}}_k(S = j^*|X^*_{k,1}), \ldots, \hat{\mathbb{P}}_k(S = j^*|X^*_{k,m}) \right)$ and $\hat{\pi}_k = \|\boldsymbol{\eta}^m_{j^*,k}(X^*_{k,\cdot})\|_\infty$, as defined in Lemma 4.4. Then, by a simple plug in to get $\hat{C}_{1,k} = \frac{\hat{\pi}_k + |\mathcal{D}| - 2}{|\mathcal{D}|\hat{\pi}_k - 1}$.
**Step 3: Averaging:** we then estimate $C_1$ using $\hat{C}_1$, computed as $\hat{C}_1 = \frac{1}{n_1} \sum^{n_1}_{k=1} \hat{C}_{1,k}, \hat{C}_{1,k}, k \in [n_1]$.

Next, we state two assumptions used to derive Theorem 4.5 (noise rate is estimated from the data).
**Assumption A:** (Sub-exponentiality) For all $k \in [n_1]$, define $\hat{g}_k(X^*) = \hat{\mathbb{P}}_k(S = j^*|X^*)$ There exists a constant $M_g > 0$, such that $\left\| \hat{C}_{1,k} \right\|_{\psi_1} = \left\| \min_{i \in [m]} \frac{\hat{g}_k(X^*_{k,i}) + |\mathcal{D}| - 2}{|\mathcal{D}|\hat{g}_k(X^*_{k,i}) - 1} \right\|_{\psi_1} \leq M_g$ for all $k \in [n_1]$, where $\| \cdot \|_{\psi_1}$ is the sub-exponential norm: $\|X\|_{\psi_1} = \inf\{t > 0 | \mathbb{E}[e^{X/t}] \leq 2\}$.

**Assumption B:** (Nearly Unbiasedness) For all $k \in [n_1]$, $\hat{C}_{1,k}$ is a "nearly" unbiased estimator of $C_1$, namely $\left| \mathbb{E}[\hat{C}_{1,k}] - C_1 \right| < \theta$ for all $k \in [n_1]$, where $\theta > 0$.

With the above procedure and assumptions, we derive the following theorem:

**Theorem 4.5.** *For any $\delta \in (0, \frac{1}{3})$, $C_1 = \frac{\pi + |\mathcal{D}| - 2}{|\mathcal{D}|\pi - 1}$, denote $VC(\mathcal{F})$ as the VC-dimension of the hypothesis class $\mathcal{F}$, and $K$ be some constant that depends on $VC(\mathcal{F})$. If Assumption A, B, and Lemma 4.4 hold, let $f = \{f_k\}^{|\mathcal{D}|}_{k=1}$ where $f_k \in \mathcal{F}$ and let $L : Y \times Y \to \mathbb{R}_+$ be a loss function bounded by some constant $M$. Denote $k^* \leftarrow \arg\max_k |\hat{\mathcal{R}}^{LDP}(f_k)\hat{\mathbb{P}}(D = k) - \mathcal{R}^{LDP}(f_k)\mathbb{P}(D = k)|$, if $n \geq \frac{8\ln(\frac{|\mathcal{D}|}{\delta})}{\min_k \mathbb{P}(S=k)}$, $n_1 \geq \frac{1}{c(\tilde{\epsilon} - \theta)^2}(M_g + \frac{C_1 + \theta}{\ln 2})^2 \ln(\frac{2}{\delta})$, and $M_g + \frac{C_1 + \theta}{\ln 2} > \tilde{\epsilon} > \theta$, where $c$ is an absolute constant, then with probability $1 - 3\delta$:*

$$\hat{\mathcal{R}}^{LDP}(f) \leq \mathcal{R}(f^*) + K\sqrt{\frac{VC(\mathcal{F}) + \ln(\frac{\delta}{2})}{2n}} \frac{2(C_1 + \tilde{\epsilon})M|\mathcal{D}|}{\mathbb{P}(S = k^*)}.$$

**Remark 1:** $\hat{C}_1$ is not explicitly shown in the bound, but it is a vital element that connects assumptions and estimation procedure. Its randomness is absorbed in $\tilde{\epsilon}$ (see proof in Appendix C.4).

**Remark 2:** As $n_1$ increases, $\hat{C}_1$ is more accurate, as it is the average of $n_1$ independent variables, resulting in a tighter bound. However, blindly choosing a large $n_1$ is not recommended, since Assumption A will not hold if $m = \frac{n}{n_1}$ is too small. Some light tuning may help select $n_1$ in practice.

**Remark 3:** Generally speaking, the bound is more adversely affected by the underestimation of $\pi$. Note that the parameter that significantly influences in the error bound is $\frac{1}{\pi - 1/|\mathcal{D}|}$. Hence, when $\pi$ is close to $\frac{1}{|\mathcal{D}|}$, an underestimation of $\pi$ can be far more detrimental than an overestimation (especially for $\hat{\pi} \leq \frac{1}{|\mathcal{D}|}$). Further, we provide an empirical study on the effect of estimation error in Section 5.2.

# 5 EXPERIMENTS & RESULT

We evaluate the performance of our proposed method using two datasets, demonstrating that the experiment results are in support of our theories. Building on these findings, we further provide practical guidelines for implementing our method under various conditions.

We evaluate our method in a regression task (MSE loss) with the US Health Insurance dataset, as well as in classification tasks (Cross-Entropy loss) with an Auto Insurance dataset. While this section presents the results from the regression task involving the US Health Insurance data, results from the classification task with the Auto Insurance dataset, which exhibit similar patterns, are in Appendix F.

The US Health Insurance dataset contains 1338 observations, 6 features, and 1 response. In our experiment, we select sex (with values "Male" and "Female") as the sensitive attribute $D$. The privatized sensitive attribute $S$ is generated under different privacy levels using a set of $\epsilon$'s by Definition 4.1. $D$ serves as the performance benchmark and is masked in all other settings, with all results computed as the mean across five seeds. We conduct experiments in two scenarios: 1) when noise rates are known and 2) when noise rates are unknown. To investigate how a transformation $T$ may affect the performance of our method, we consider a transformation $T(X) = \tilde{X}$ obtained via supervised learning (as shown in Example 4.1). Other transformations, such as grouping, and discretization are also commonly applied in insurance pricing.

In both scenarios, we let the hypothesis class $\mathcal{F}$ be the class of linear models across all settings, as this aligns with the transparency requirements prevalent in insurance pricing. We ran our algorithm with three pre-defined $\pi$'s, namely $0.8, 0.7$, and $0.6$, to assess how varying noise levels impact our method's performance in each scenario. Additionally, we created subsets of the original dataset to examine how sample size influences our method's efficacy. In Scenario 2, we further compared performance using three different $n_1$ values, namely $1, 2$, and $4$, to validate our findings in Theorem 4.5. To obtain the discrimination-free price $h^*$, we choose $P^*(d)$ to be the empirical marginal distribution of $D$. In all figures in this section, while the blue curves (Best-Estimate, as in Definition 3.1), the orange curve (MPTP), and the rest (MPTP-LDP) were all obtained using a logistic regression, different score functions were used. A conventional score function is used to obtain Best-Estimate and group-specific score functions (as defined in Eq. (1)) were used to obtain MPTP and MPTP-LDP.

Since the main challenge is estimating $\mu(X, D)$ when $D$ is inaccessible, we focus on presenting the results for this estimation. However, results for $h^*(X)$ in both scenarios are included in Appendix F

## 5.1 SCENARIO 1: KNOWN NOISE RATE

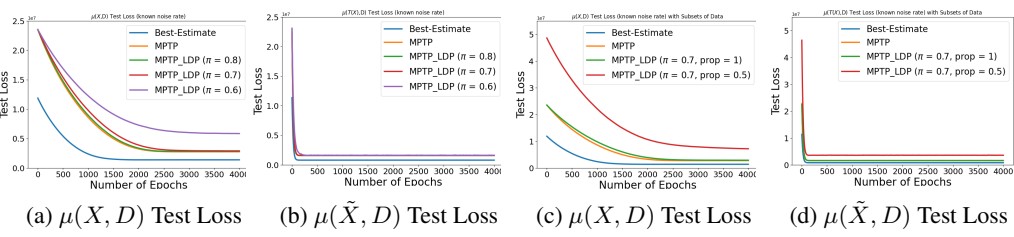

(a) $\mu(X, D)$ Test Loss    (b) $\mu(\tilde{X}, D)$ Test Loss    (c) $\mu(X, D)$ Test Loss    (d) $\mu(\tilde{X}, D)$ Test Loss

Figure 2: Test Loss for Scenario 1 (fixed sample size: (a)(b), fixed noise rate: (c)(d))

Figure 2a 2b show how the noise rate affects loss approximation with a fixed sample size, where we observe a huge difference in terms of convergence rate and robustness against noise rate. This is expected since $\tilde{X}$ already incorporates some information from the response $Y$. Thus, the impact of noise perturbations is diminished, leading to increased robustness and faster convergence. Additionally, we note that a higher noise rate generally results in a larger error gap when the sample size remains fixed, which is consistent with our findings in Theorem 4.3. From a practical point of view, when the sample size is sufficiently large, an appropriate transformation can be beneficial in scenarios where 1) the noise rate is high, 2) computing resources are limited, or 3) a tight error gap is essential.

In Figures 2c 2d, we examine the effect of sample size on loss approximation by randomly creating a subset of the full dataset that contains half of its observations. We then conduct the same experiment on both the full dataset (green curve) and the subset (red curve). Our findings reveal a marked difference in convergence rates between $X$ and $\tilde{X}$. Furthermore, for any fixed noise rate, a larger sample size generally leads to a lower test loss, irrespective of the transformation applied. This observation is consistent with the results presented in Theorem 4.3.

## 5.2 SCENARIO 2: UNKNOWN NOISE RATE

Similar to scenario 1, the primary distinction is that $\pi$ is replaced by an estimate $\hat{\pi}$ obtained using Lemma 4.4. To illustrate consistency with our theoretical results in Theorem 4.5, we present comparisons not only under fixed sample sizes and true noise rates but also with different $n_1$ values. To estimate $\pi$, we randomly and evenly split the full data set into $n_1$ subsets and compute $\hat{\pi}_k$ for $k = 1, \ldots, n_1$ on each subset. Averaging these estimates, we obtain $\hat{\pi} = \frac{1}{n_1} \sum_{k=1}^{n_1} \hat{\pi}_k$ for loss approximation in Figure 3 and Figure 4. We first present the empirical results regarding the effect of noise rate on loss approximation with a fixed sample size:

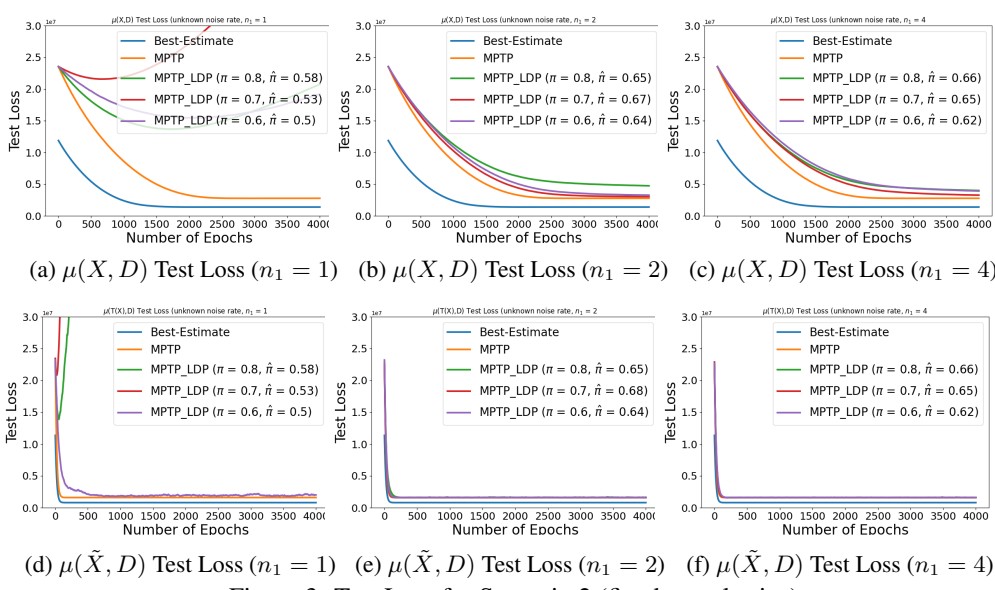

(a) $\mu(X, D)$ Test Loss ($n_1 = 1$)    (b) $\mu(X, D)$ Test Loss ($n_1 = 2$)    (c) $\mu(X, D)$ Test Loss ($n_1 = 4$)

(d) $\mu(\tilde{X}, D)$ Test Loss ($n_1 = 1$)    (e) $\mu(\tilde{X}, D)$ Test Loss ($n_1 = 2$)    (f) $\mu(\tilde{X}, D)$ Test Loss ($n_1 = 4$)

Figure 3: Test Loss for Scenario 2 (fixed sample size)

We observe similar patterns in terms of convergence rate, robustness against noise, and the implications of applying transformation, as discussed in Scenario 1 with a fixed sample size. In addition to aligning with Theorem 4.5, we note that increasing $n_1$ leads to more accurate estimates of $\pi$, resulting in improved loss approximations. This insight is one of the key practical takeaways from Theorem 4.5. However, we emphasize again that a larger $n_1$ does not always guarantee a more precise estimation, therefore, some tuning may be necessary to select an optimal $n_1$ in practice.

In Figure 3a 3d, we observe that the curves for lower true noise rates (i.e. $\pi = 0.8$ and $\pi = 0.7$) surprisingly fail to converge. Theorem 4.5 suggests that the error gap is controlled by the quality of the estimation of $\hat{\pi}$. While the value of $n_1$ offers some understanding of the robustness of $\hat{\pi}$, more specific insights remain elusive. Let us keep this issue in mind for now and examine the results regarding the effect of sample size on loss approximation for a fixed noise rate ($\pi = 0.8$) in Figure 4.

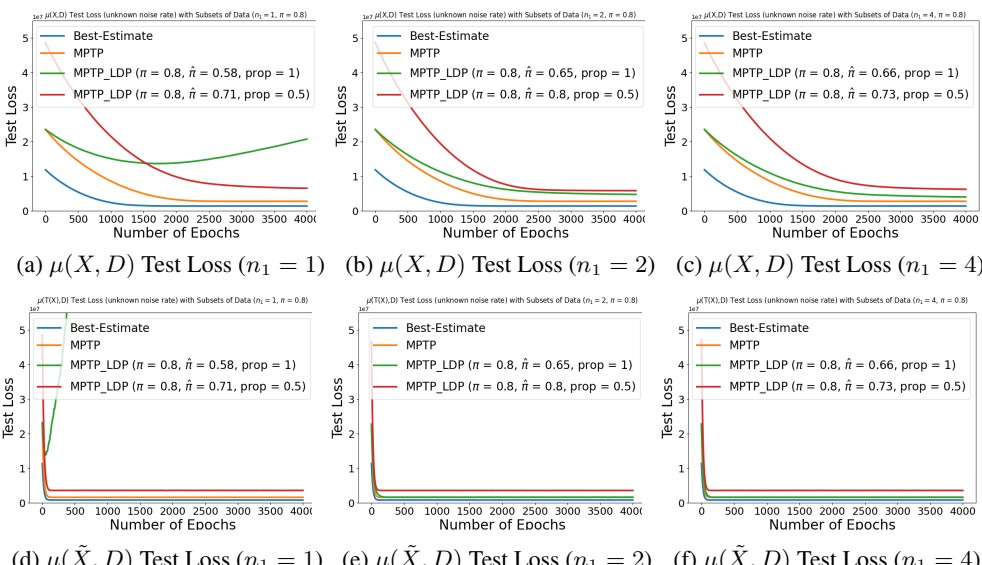

(a) $\mu(X, D)$ Test Loss ($n_1 = 1$)    (b) $\mu(X, D)$ Test Loss ($n_1 = 2$)    (c) $\mu(X, D)$ Test Loss ($n_1 = 4$)

(d) $\mu(\tilde{X}, D)$ Test Loss ($n_1 = 1$)    (e) $\mu(\tilde{X}, D)$ Test Loss ($n_1 = 2$)    (f) $\mu(\tilde{X}, D)$ Test Loss ($n_1 = 4$)

Figure 4: Test Loss for Scenario 2 (fixed noise rate: $\pi = 0.8$)

We observe patterns similar to those in scenario 1, for a fixed noise rate and transformation, a larger sample size leads to a smaller error gap. However, a closer examination of Figure 2c and Figure 4a 4b 4c, reveals that while the green curve (full data) and red curve (half data) converge empirically when the true $\pi$ is known, as shown in Figure 2c, 2d, they fail to converge when $\pi$ is unknown and estimated with $n_1 = 1$, as indicated in Figure 4a. In contrast, for $n_1 = 2$ and $n_1 = 4$, both

curves show empirical convergence. By combining insights from Figure 3 and Figure 4, we identify two key points: 1) estimation error tolerance is highly linked to $\pi$, 2) underestimation of $\pi$ tends to cause issues with empirical convergence. This motivates us to further investigate the impact of underestimation and overestimation of $\pi$ on the empirical performance of our algorithm.

### 5.3 EMPIRICAL STUDY ON THE IMPACT OF NOISE RATE ESTIMATION ERROR

Building on our observations in Section 5.2, we present our findings on the effect of estimation error for $\pi$ on the empirical performance of our algorithm. We examine how both underestimation and overestimation for $\pi$ influence the algorithm's performance under balanced and imbalanced distribution for privatized sensitive attributes $S$ by introducing pre-defined estimation errors. For imbalanced distributions, we sampled subsets from the full dataset. We present results for the balanced case below. The imbalanced case is deferred to Appendix E, as similar patterns were observed.

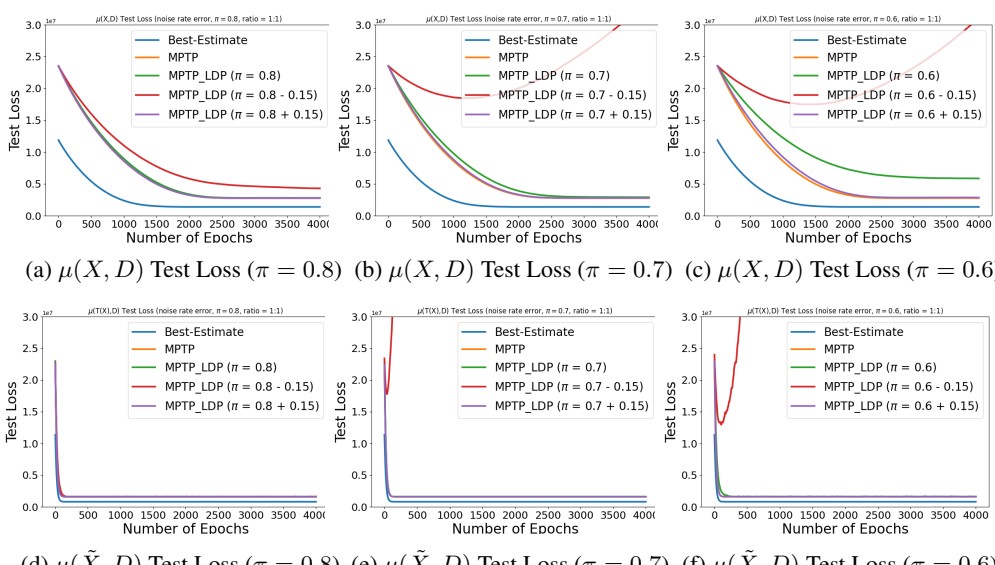

(a) $\mu(X, D)$ Test Loss ($\pi = 0.8$) (b) $\mu(X, D)$ Test Loss ($\pi = 0.7$) (c) $\mu(X, D)$ Test Loss ($\pi = 0.6$)

(d) $\mu(\tilde{X}, D)$ Test Loss ($\pi = 0.8$) (e) $\mu(\tilde{X}, D)$ Test Loss ($\pi = 0.7$) (f) $\mu(\tilde{X}, D)$ Test Loss ($\pi = 0.6$)

Figure 5: Test Loss for Noise Rate Estimation Error (error = $\pm 15\%$)

While estimation error invariably introduces bias in loss approximation, Figure 5 reveals that a lower noise rate is less sensitive to estimation error in terms of convergence behavior. For a sufficiently large $\pi$ (i.e. $\pi = 0.8$), even a significant estimation error (i.e. $\hat{\pi} = \pi \pm 15\%$) does not hinder convergence. However, excessively large errors may still lead to convergence failures even for a large $\pi$, as illustrated in Figure 3a, 3d, 4a, 4d. Notably, while estimation errors may cause convergence issues, underestimation proves to be far more detrimental to the convergence than overestimation, as shown in Figure 5. This intriguing finding aligns with our insights in Theorem 4.5, which suggests a potential solution to convergence issues: introducing a small positive constant to $\hat{\pi}$ may help.

## 6 CONCLUSION

In this paper, we proposed an efficient and practical method to achieve fairness in insurance pricing within a multi-party training framework. This framework leverages a trusted third party (TTP) to handle sensitive attributes when insurers lack direct access to such information. We derived a population-equivalent risk that can be optimized using only privatized sensitive attributes, both when the privatization noise rate is known and unknown, and we provided statistical guarantees for each scenario. Our theoretical findings reveal how the sample size and noise rate influence the error gap, offering practical guidelines for implementing the method. In our experiments, we validate our theoretical results and show that our method achieves fair pricing effectively regardless of known and unknown noise rate. The main limitation of our work is that the risk bound in Theorem 4.3 and Theorem 4.5 may be less informative in certain scenarios. For instance, regulatory transparency requirements might prevent insurers from applying dimension-reduction techniques to high-cardinality sensitive attributes. Future work could extend the framework to accommodate continuous sensitive attributes and adapt it to other fields with similar regulatory constraints.

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
