# OpenReview forum: "Discrimination-free Insurance Pricing with Privatized Sensitive Attributes"
_ICLR.cc/2025/Conference — Submitted to ICLR 2025_

### Official Review · Reviewer_2RA7 · 2024-11-02

**Soundness:** 3
**Presentation:** 3
**Contribution:** 3
**Rating:** 6
**Confidence:** 3

**Summary:**

This paper develops a method for learning a "discrimination-free" insurance pricing policy when sensitive attributes are private, and only noisy forms of these attributes are available to a trusted third party. Theoretical results are presented for the case where the privatization approach is a) fully known, b) has unknown noise rates. The method is applied to health and auto insurance datasets.

**Strengths:**

Though I'm not an expert in privacy-preserving ML, this paper's theoretical results are probably of interest to people in that field.

After revisions, this paper sufficiently discusses the practical implications of the proposed approach to insurance markets and customers in those markets.

**Weaknesses:**

UPDATE: After discussion and revisions, I believe the paper has adequate answers to these questions. Though I'd like to see the substantive impacts of the proposed approach get even more attention in the paper, I think the authors have done enough to merit acceptance.

The experiments and evaluation section in this paper claims to show that the method "achieves fair pricing effectively", but it answers none of the questions that would allow us to determine if such pricing is fair, effective, or desirable.

* How much do men and women pay for insurance after this method is applied?
* How does this compare to the benefit they receive from insurance payouts?
* Which other subgroups benefit or are made worse off by this method?
* If insurance is under/overpriced it could lead to adverse selection, where, for example, high-risk male drivers buy more insurance because it’s cheap, increasing premiums for everyone else. It could even lead to people driving more dangerously at the margin because they know that an incident won’t increase their premiums much. Is there risk of adverse selection or other negative equilibrium effects from using this pricing?

This paper paper is clearly trying to develop a method for pricing that can be used by real insurers. Unfortunately, though, it treats insurance pricing as an exercise in privacy math, and not as an input to a crucially important product for people's physical and financial health.

**Questions:**

Please see my questions on the important practical implications of this method above. In addition:

Convergence in a linear model with 6 features requiring thousands of epochs seems very slow to me - what's causing this?

What's the reason for prioritizing convergence rate in the experiments? This doesn't seem like an important property of insurance pricing algorithms (unless, of course, the model diverges), since the time and expense of training the model will likely be very small compared to the revenue and expenses of actually delivering insurance.

---

> ### Author Response · Authors · 2024-11-20
> **Response to Reviewer 2RA7 (Part 1)**
>
> Dear Reviewer 2RA7:
>
> Thank you for taking the time to review our paper and for providing thoughtful and constructive feedback. We have carefully considered your comments and incorporated them into our revisions.
>
> Below, you will find our detailed responses to each of your questions and concerns.
>
> $\textbf{Q1:}$ "The experiments and evaluation section in this paper claims to show that the method "achieves fair pricing effectively", but it answers none of the questions that would allow us to determine if such pricing is fair, effective, or desirable."
>
> $\textbf{A1:}$ To clarify, our approach follows a structured workflow: first, fairness must be defined, and second, methods are proposed to determine a price that satisfies this fairness definition. This workflow aligns with how fairness is typically investigated in other disciplines. In our case, we adopt the fairness definition from Lindholm (2021) to define what constitutes a fair price (Definition 3.3 in the paper). The focus of our paper is to propose a framework for determining a fair price when sensitive attributes are noised. To address your question, the premium determined by the proposed method is fair according to the fairness definition from Lindholm (2021).
>
> $\textbf{Q1.1:}$ How much do men and women pay for insurance after this method is applied?
>
> $\textbf{A1.1:}$ To clarify, our work focuses on situations where the use of sensitive attributes in pricing is prohibited due to regulatory constraints. For example, Directive 2004/113/EC (“Gender Directive”) issued by the Council of the European Union prohibits insurance companies in the EU from using gender as a rating factor for pricing insurance products. Under such a regulation, if gender is not allowed to be used in pricing, our proposed method ensures that an otherwise identical woman and man will pay the same premium.
>
> Additionally, we emphasize that the proposed method enforces individual fairness rather than group fairness. This means we are not suggesting that all women and all men should pay the same premium, but rather that individuals with identical risk profiles, regardless of gender, will be treated equally. In our framework, insurers are still able to use non-sensitive attributes to assess and price risk, thus ensuring actuarial fairness, where premiums are commensurate with the expected costs associated with each individual.
>
> $\textbf{Q1.2:}$ "How does this compare to the benefit they receive from insurance payouts?"
>
> $\textbf{A1.2:}$ As we have previously noted, the fairness enforced by our proposed method operates at the individual risk level. This ensures that each policyholder pays a premium commensurate with their risk profile, as defined by the non-sensitive attributes used by the insurer. Specifically, for male and female policyholders, their premiums are proportional to their riskiness as reflected by their respective risk classes. Consequently, for homogeneous risks within a given risk class, the total premiums collected are expected to align closely with the total insurance payouts, maintaining consistency with actuarial fairness principles.
>
> $\textbf{Q1.3:}$ "Which other subgroups benefit or are made worse off by this method?"
>
> $\textbf{A1.3:}$ No other subgroups are advantaged or disadvantaged by the proposed method. A toy example is that, given the risk profile $X_1$, the break-even premium $\mu(X_1, D = 0) > \mu(X_1, D = 1)$, however for another different risk profile $X_2$, the break-even premium $\mu(X_2, D = 0) < \mu(X_2, D = 1)$. Then, by paying the discrimination-free premium (Definition 3.3), it is hard to define which subgroup is worse off, since in each subgroup, some individuals are made better off and some are made worse off.
>  As we have previously explained, insurers still classify and differentiate risks based on non-sensitive attributes permitted by regulators. The primary role of our method is to ensure that policyholders with identical risk profiles, as determined by these non-sensitive attributes, are not discriminated against based on sensitive characteristics. This approach upholds fairness by ensuring that premiums are commensurate with the expected costs for each risk class, regardless of sensitive attributes.

---

> ### Author Response · Authors · 2024-11-20
> **Response to Reviewer 2RA7 (Part 2)**
>
> $\textbf{Q1.4:}$ "If insurance is under/overpriced it could lead to adverse selection, where, for example, high-risk male drivers buy more insurance because it’s cheap, increasing premiums for everyone else. It could even lead to people driving more dangerously at the margin because they know that an incident won’t increase their premiums much. Is there risk of adverse selection or other negative equilibrium effects from using this pricing?"
>
> $\textbf{A1.4:}$ Thank you for your insightful comments. You raise a critical issue tied to the fundamental problem of information asymmetry in insurance, where insurers rely on risk classification as a key tool to differentiate and manage risks. This concern underscores why the concept of fairness holds unique importance in the insurance context and why it differs significantly from its application in other domains.
>
> First, in our proposed framework, insurers use non-sensitive attributes for risk classification, which helps address adverse selection concerns. Additionally, the framework supports the use of experience rating, allowing insurers to adjust premiums based on a policyholder’s claim history, thereby mitigating potential moral hazard effects.
>
> Second, we understand your concern that prohibiting the use of sensitive attributes in pricing—after risk classification based on non-sensitive attributes—could introduce potential adverse selection. This is a valid consideration. However, determining whether regulators should enforce such prohibitions involves a broader investigation into the trade-offs between fairness and market efficiency. While this question is outside the scope of our current study, it is a critical topic we are actively exploring in a separate paper.
>
> $\textbf{Q2:}$ "This paper is clearly trying to develop a method for pricing that can be used by real insurers. Unfortunately, though, it treats insurance pricing as an exercise in privacy math, and not as an input to a crucially important product for people's physical and financial health."
>
> $\textbf{A2:}$ The primary aim of this paper is to develop a practical framework for pricing insurance products in scenarios where sensitive attributes, those restricted by regulatory constraints, are involved. We are sorry to hear that you perceive our work as overly focused on privacy mathematics. However, we believe our framework addresses a significant regulatory and operational challenge and can be readily implemented using existing protocols between insurers and third parties.
>
> We understand that your assessment may stem from a misperception of our approach, and we hope our responses to your remarks provide greater clarity regarding the scope and focus of our work. We trust this clarifies how the proposed framework contributes to both theoretical advancements and real-world applicability, and we hope you now view it as a valuable contribution to ICLR 2025.
>
> $\textbf{Q3:}$ "Convergence in a linear model with 6 features requiring thousands of epochs seems very slow to me - what's causing this?"
>
> $\textbf{A3:}$ Regarding the convergence rate, we agree that the speed might appear slow, especially given the linear model with only six features. However, it's important to note that the convergence speed is not the main focus of our study. We do acknowledge that fine-tuning hyperparameters, such as learning rates, can certainly improve convergence. In our experiments, we specifically tuned the parameters to favor the linear model's performance. For example, in the comparison between using original and transformed attributes, we ensured that the learning rate for the linear model using $X$ was set much higher than that for the model using $\tilde{X}$.
>
> In addition, to accommodate the large datasets in real-world applications, we specifically employed a data loader, mini-batch training, and an SGD-based (Adam) optimizer in our experimentation. With mini-batch and Adam used in all experiments, and given that the transformed attributes (i.e. $\tilde{X}$) already contain much information about $Y$ through the construction of the transformation of $T$, in general, we do expect to observe:
> 1) a sharp difference in convergence rate between $X$ and $\tilde{X}$.
> 2) A much slower convergence rate for the linear model in an absolute value sense.
>
> Additionally, we have thoroughly checked our code and confirmed the correctness of our implementation. As part of our commitment to transparency, we have also submitted the data and code along with the paper for replication and further investigation by readers.

---

> ### Author Response · Authors · 2024-11-20
> **Response to Reviewer 2RA7 (Part 3)**
>
> $\textbf{Q4:}$ "What's the reason for prioritizing convergence rate in the experiments? This doesn't seem like an important property of insurance pricing algorithms (unless, of course, the model diverges), since the time and expense of training the model will likely be very small compared to the revenue and expenses of actually delivering insurance."
>
> $\textbf{A4:}$ To clarify, our experiments did not prioritize convergence rate as a key metric. We suspect your conclusion stems from the comparison between using original versus transformed non-sensitive attributes in our numerical experiments. In these experiments, the transformed attributes are learned by the insurer based on insurance loss outcomes, which naturally leads to faster convergence compared to the case of using original attributes. However, this outcome is a byproduct of the transformation process, not the focus of our analysis.
>
> The motivation for using transformations, denoted as $T$, is rooted in practical, real-world considerations rather than computational efficiency. For instance, insurers may wish to safeguard their pricing strategies when collaborating with a trusted third party (TTP). A straightforward transformation can effectively obscure original rating variables for this purpose. Alternatively, a more complex transformation could be employed to improve risk assessment, leveraging auxiliary data that is more extensive than the dataset shared with the TTP for a specific pricing task.

---

> > ### Comment · Reviewer_2RA7 · 2024-11-23
> >
> > Thanks for patient explanation here. The key thing I'm confused about is represented by these two quotes:
> >
> > > but rather that individuals with identical risk profiles, regardless of gender, will be treated equally.
> >
> > > No other subgroups are advantaged or disadvantaged by the proposed method.
> >
> > Suppose X only includes the number of tickets for traffic violations the driver has received in the past 5 years. And suppose men drive more recklessly and have more tickets on average. Thus, gender can be inferred from X. Because of this, the paper suggests that it would be unfair to price insurance according to E[Y|X] (Line 177). Alternative pricings h(X) != E[Y|X] would either increase premiums for high-risk people or low-risk people, benefitting (on average) either female or male drivers and disadvantaging the other group. What am I missing here?

---

> > > ### Author Response · Authors · 2024-11-24
> > > **Response to Reviewer 2RA7 (Follow-up 1, Part 1)**
> > >
> > > Dear Reviewer 2RA7,
> > >
> > > Thank you for your follow-up comments. We appreciate the opportunity to provide further clarification regarding our proposed rating method. It appears that the confusion is due to the distinction between the concepts of “individual fairness” and “group fairness”.  Our work is grounded in the principle of “individual fairness,” as opposed to “group fairness,” which is not suitable for ratemaking purposes.
> > >
> > > Let us elaborate using the example you provided. Suppose X represents the number of tickets for traffic violations. To keep it simple, let’s assume there are only two possibilities: $X=0$ (low) and $X=1$ (high). Suppose $D$ represents gender with two possibilities, $D=M$ (male) and $D=F$ (female). In this context:
> > > 1. Group fairness means: The average premium for the group of males (which includes both $X=0$ and $X=1$) is equal to the average premium for the group of females (which also includes both $X=0$ and $X=1$). This fairness is in the aggregate sense, in that, the “average premium per person” is calculated across both $X=0$ and $X=1$ for each of the male and female groups.
> > > 2. Individual fairness means: For the sub-population of $X=0$, the average premium for the group of males is equal to the average premium for the group of females. Similarly, for the sub-population of $X=1$, the average premium for the group of males is equal to the average premium for the group of females. This fairness applies at the individual level, in that, the “average premium per person” is calculated for male and female groups that are otherwise identical, i.e. same $X$.
> > >
> > > Further assume that men drive more recklessly and have more tickets on average. This implies $X$ and $D$ are correlated, meaning $D$ can be (partially) inferred from $X$. It is important to note that a perfect correlation between $X$ and $D$ is both trivial and unrealistic. This means that $X=0$ would always correspond to $D=F$ (female), and $X=1$ would always correspond to $D=M$ (male). In this case, the concept of individual fairness is not applicable because the sub-groups of $X=0$ and $X=1$ no longer exist for either male ($D=M$) or female ($D=F$).
> > >
> > > Given the above setting, our paper focuses on insurance pricing that satisfies individual fairness in cases where $D$ and $X$ are correlated. As you have correctly pointed out, given correlation between $D$ and $X$, the average premium for males is not the same as those for females, which concerns the concept of group fairness. This highlights the fact that individual fairness does not necessarily guarantee group fairness, a result has been established in existing literature. For instance, see Dwork et al (2011)[1] and Lindholm et al (2022)[2].
> > >
> > > [1]​​ Cynthia Dwork, Moritz Hardt, Toniann Pitassi, Omer Reingold, and Richard Zemel. Fairness through
> > > awareness. In Proceedings of the 3rd Innovations in Theoretical Computer Science Conference,ITCS ’12, pp. 214–226, New York, NY, USA, 2012. Association for Computing Machinery.
> > > ISBN 9781450311151. doi: 10.1145/2090236.2090255. URL https://doi.org/10.1145/
> > > 2090236.2090255
> > >
> > > [2] Mathias Lindholm, Ronald Richman, Andreas Tsanakas, and Mario V. W ¨uthrich. A discussion of discrimination and fairness in insurance pricing, 2022b.

---

> > > ### Author Response · Authors · 2024-11-24
> > > **Response to Reviewer 2RA7 (Follow-up 1, Part 2)**
> > >
> > > To be more specific, our proposed method guarantees a discrimination-free premium for the groups of $X=0$ and $X=1$ separately, achieving fairness at an individual level. This is not a trivial task when $X$ and $D$ are correlated. To clarify, let us make the example more concrete.
> > >
> > > Let’s assume among the entire population of drivers, $50\%$ are male and 50% are female. That is, $P(D=M) = 0.5$ and $P(D=F) = 0.5$. Further, the correlation between D and X is modeled by the assumption:
> > > 1. Among male drivers ($D=M$), $10 \\%$ have $X=0$, and the rest $90\\%$ have $X=1$. That is, $P(X=0|D=M) = 0.1$ and $P(X=1|D=M) = 0.9$.
> > > 2. Among female drivers ($D=F$), $90 \\%$ have $X=0$, and the rest $10\\%$ have $X=1$. That is, $P(X=0|D=F) = 0.9$ and $P(X=1|D=F) = 0.1$.
> > >
> > > Using Bayes’ theorem, we can derive the following probabilities:
> > > 1. $P(D=M|X=0$) = $(0.1 \times 0.5) / (0.1 \times 0.5 + 0.9 \times 0.5) = 0.1$
> > > 2. $P(D=F|X=0) = 1 - 0.1 = 0.9$
> > > 3. $P(D=M|X=1) = (0.9 \times 0.5) / (0.1 \times 0.5 + 0.9 \times 0.5) = 0.9$
> > > 4. $P(D=F|X=1) = 1 - 0.9 = 0.1$
> > >
> > > For pricing purposes, our goal is to determine the premium for the two classes $X=0$ and $X=1$. Since regulators prohibit the use of $D$ (gender) in rating, a male and a female policyholder with the same $X$ value will be assigned the same premium.
> > >
> > > Under current ratemaking practices, the "expected cost" serves as the basis for determining the pure premium. This is referred to the unawareness premium (Definition 3.2 in the paper), calculated as:
> > >
> > > 1. $E[Y|X=0] = E[Y|X=0,D=M] * P(D=M|X=0) + E[Y|X=0,D=F] * P(D=F|X=0) = E[Y|X=0,D=M] * 0.1 + E[Y|X=0,D=F] * 0.9$
> > > 2. $E[Y|X=1] = E[Y|X=1,D=M] * P(D=M|X=1) + E[Y|X=1,D=F] * P(D=F|X=1) = E[Y|X=1,D=M] * 0.9 + E[Y|X=1,D=F] * 0.1$
> > >
> > > In this case, while individuals with the same $X$ value (e.g., $X=0$ or $X=1$) pay the same premium regardless of gender ($D$), gender is implicitly discriminated against because $D$ and $X$ are correlated.
> > >
> > > To better understand this, suppose $E[Y|X=0,D] = E[Y|X=1,D$] for $D=M$ and $F$, i.e. given gender, $X$ is not predictive of cost. Above calculation (unawareness premium) suggests that $E[Y|X=0] <  E[Y|X=1]$. In other words, even though $X$ is not predictive of $Y$, the insurer charges different premiums for $X=0$ and $X=1$. This aligns with your observation that gender can be inferred through $X$. Consequently, the average premium for male policyholders is higher than the average premium for female policyholders.
> > >
> > > To address this issue of implicit unfairness, we employ the discrimination-free premium, Definition 3.3 in the paper, where the premium for each class ($X=0$ and $X=1$) is determined based on $P^*(D)$ instead of $P(D|X)$. Let's choose $P^*(D = M) = P^*(D = F) = 0.5$. Then, the discrimination-free premiums are calculated as:
> > >
> > > 1. $h^*(X=0) = E[Y|X=0,D=M] * P^*(D=M) + E[Y|X=0,D=F] * P^*(D=F) = E[Y|X=0,D=M] * 0.5 + E[Y|X=0,D=F] * 0.5$
> > > 2. $h^*(X=1) = E[Y|X=1,D=M] * P^*(D=M) + E[Y|X=1,D=F] * P^*(D=F) = E[Y|X=1,D=M] * 0.5 + E[Y|X=1,D=F] * 0.5$
> > >
> > > Now consider the same scenario $E[Y|X=0,D]=E[Y|X=1,D]$ for $D= M$ and $F$, i.e. given gender, $X$ is not predictive of cost. The discrimination-free premium implies that $h^*(X=0) = h^*(X=1)$. In other words, the insurer charges the same premium for $X=0$ and $X=1$. As a result, the average premium per male is the same as the average premium per female.
> > >
> > > Of course it is more sensible to assume $E[Y|X=0,D] <  E[Y|X=1,D]$ for $D=M$ and $F$, i.e. given gender, $X$ is predictive of $Y$. In this case, both male and female still pay the same premium, i.e. $h^*(X=0)$ when $X=0$, and $h^*(X=1)$ when $X=1$. However, in aggregate (when combining $X=0$ and $X=1$), the average premium per male is higher than the average premium per female. As mentioned before, this example underscores the important distinction between individual and group fairness. And in our context, individual fairness does not guarantee group fairness.
> > >
> > > We hope our response has provided additional clarity on the scope and contributions of our work. If you find our explanations satisfactory, we kindly ask you to consider adjusting your overall rating for our paper.

---

> ### Comment · Reviewer_2RA7 · 2024-11-24
>
> Thanks for the reply. To quickly ask for further clarification:
>
> > To address this issue of implicit unfairness,
>
> Unfairness on what basis? You said your work is concerned with individual fairness, and the unawareness premium appears to satisfy your definition of individual fairness above.
>
> > i.e. given gender, X is not predictive of cost.
>
> This seems like a very strange assumption. Surely it's vastly more likely that once we know someone's driving behavior their gender is not predictive of cost? I guess if $ Y \perp D | X $ then then the discrimination-free premium matches the unawareness premium? So if we stipulate that "true" risk/cost is not a function of gender, the discrimination-free premium differs from the unawareness premium only to the extent that omitted variables bias causes us to measure a spurious difference: $E[Y|X, D=M] - E[Y|X, D=F] \neq 0$?

---

> > ### Author Response · Authors · 2024-11-24
> > **Response to Reviewer 2RA7 (Follow-up 2)**
> >
> > Dear Reviewer 2RA7,
> >
> > Thanks for your timely response.
> >
> > $\textbf{Q1:}$ "Unfairness on what basis? You said your work is concerned with individual fairness, and the unawareness premium appears to satisfy your definition of individual fairness above."
> >
> > $\textbf{A1:}$ Yes, you are correct if being individually fair is all we are concerned about, then the premium itself is indeed individually fair since for any given $X$, regardless of the value $D$ takes, the premiums are the same. However, in the fair insurance pricing literature, the use of $P(D|X)$ in the calculation of unawareness premium is identified as a source of indirect discrimination and, thus, should be avoided. Therefore, from our understanding, the discrimination-free premium (Definition 3.3) is stronger than simply being individually fair. It further removes the indirect discrimination from the use of $P(D|X)$.
> >
> > $\textbf{Q2: }$ "This seems like a very strange assumption. Surely it's vastly more likely that once we know someone's driving behavior their gender is not predictive of cost? I guess if $Y \perp D \mid X$ then then the discrimination-free premium matches the unawareness premium? So if we stipulate that "true" risk/cost is not a function of gender, the discrimination-free premium differs from the unawareness premium only to the extent that omitted variables bias causes us to measure a spurious difference: $E[Y \mid X, D=M]-E[Y \mid X, D=F] \neq 0$ ?"
> >
> > $\textbf{A2:}$ Yes, if $Y \perp D | X$ holds, then we essentially have the following:
> >
> > 1. The unawareness premium: $\mu(X) = \int_d \mu(X,d) d\mathbb{P}(d|X) = \int_d \mu(X) d\mathbb{P}(d|X) = \mu(X)$
> > 2. The discrimination-free premium: $h^*(X) = \int_d \mu(X,d) d\mathbb{P}^*(d) = \int_d \mu(X) d\mathbb{P}^*(d) = \mu(X)$
> >
> > These two premiums are matched as you pointed out even when $D \perp X$ does not hold. However, $Y \perp D | X$ may not hold in practice, making $D \perp X$ necessary to ensure an unawareness premium does not introduce any indirect discrimination.
> >
> > Your last statement is also correct. In the case of having omitted variables (which is almost always the case in practice since we never know what the true model is), The discrimination-free premium still gives us premiums that are free from direct, and indirect discrimination.

---

> > > ### Comment · Reviewer_2RA7 · 2024-11-24
> > >
> > > Ok great, this is becoming clearer to me (and more defensible).
> > >
> > > Since the goal is to support real-world insurance pricing, I think my original concern about how benefits and costs are reallocated by the discrimination-free premium still stand. In our running example, if men remain riskier than women conditional on X (perhaps because of omitted variables) then the discrimination-free premium is effectively female drivers subsidizing male drivers, right? I still think it's important to know the magnitude (if any) of these effects in real markets.
> > >
> > > I'll revise my score upwards and consider raising it further if this concern can be addressed.

---

> ### Author Response · Authors · 2024-11-25
> **Response to Reviewer 2RA7 (Follow-up 3)**
>
> Dear Reviewer 2RA7,
>
> Thank you for your prompt reply, and more importantly, for your insightful question. Consider the scenario: presumably because of omitted variables, Y (insurance claims) and D (gender) are not independent, conditional on X (rating variables). In our running example, where men remain riskier than women, as you correctly pointed out, under the discrimination-free premium, female drivers effectively subsidize male drivers.
>
> We fully agree with you that understanding the magnitude of these effects is important, and we believe this insight is valuable to both insurers and regulators. To address this concern, we would like to offer several thoughts:
> 1. Technically, an upper bound for the total magnitude of cross-subsidization can be established. It is important to note: while female drivers subsidize male drivers at the group level, the direction of subsidization may not be consistent for every individual. Consequently, the upper bound can be derived using a counterfactual approach, assuming for each $X$ (distinct risk profile in the portfolio): $\mu(X, D = F) = \max \\{\mu(X,D = F), \mu(X, D = M)\\}$ and $\mu(X, D = M) = \min \\{\mu(X,D = F), \mu(X, D = M)\\}$. This ensures each pair is subsidizing in the same direction. It is straightforward to verify that this represents the largest possible cross-subsidization, and that the upper bound is a function of $\mathbb{P}^*(d)$ since the discrimination-free premium $h^*(X)$ is computed via $\mathbb{P}^*(d)$.
> 2. We emphasize that a trade-off exists between fairness and cross-subsidization in real-world applications. The optimal approach involves finding a balance between these competing objectives rather than strictly enforcing fairness with no allowance for cross-subsidization mitigation. As we mentioned before, we are investigating this problem in a separate working paper, where we relax the fairness constraint to allow for a mitigation in cross-subsidization. However, we believe this analysis falls outside the scope of this conference. Therefore, in our submission to ICLR 2025, we have focused solely on the technical details of our proposed method.
> 3. Last, it is worth noting that while the subsidization is a valid concern for the proposed discrimination-free pricing method, insurers in practice often employ alternative strategies to mitigate the cross-subsidization caused by potential omitted variables. For example, in the European Union, where insurers are prohibited from using gender as a rating factor, studies have shown that telematics data, which captures driving behavior, can effectively substitute for gender in ratemaking (e.g., [1] and [2]).  In this context, telematics data can be seen as addressing the omitted variable problem in models that do not initially include such granular information. More broadly, insurers routinely gather additional data from various sources to account for potential omitted variables in their models.
>
> [1] Roel Verbelen, Katrien Antonio, Gerda Claeskens (2018). Unravelling the predictive power of telematics data in car insurance pricing. In Journal of the Royal Statistical Society: Series C.
>
> [2] Ayuso, Mercedes, Montserrat Guillen, and Jens Perch Nielsen. "Improving automobile insurance ratemaking using telematics: incorporating mileage and driver behaviour data." Transportation 46 (2019): 735-752.

---

> > ### Comment · Reviewer_2RA7 · 2024-11-25
> >
> > Will this appear in the final version of the paper? At the moment I don't think it emphasizes that a trade-off exists between using the non-discrimination premium and cross-subsidization. (I wouldn't call this a tradeoff between "fairness" and cross-subsidization since both sides of the concern are about some type of fairness.) In an earlier comment you seemed to suggest that such a tradeoff didn't exist: "No other subgroups are advantaged or disadvantaged by the proposed method."

---

> > > ### Author Response · Authors · 2024-11-26
> > > **Response to Reviewer 2RA7 (Follow-up 4)**
> > >
> > > Dear Reviewer 2RA7,
> > >
> > > Yes, in the newly revised paper, we have included a discussion on this trade-off. We agree that it involves two types of fairness, rather than a trade-off between fairness and cross-subsidization. We have labeled these as equity-based and risk-based fairness. Due to the page limit of the paper, our discussion is concise and presented as follows (please see line 178-186 on page 4, highlighted in red).
> > >
> > > “Note that if $Y\perp D|X$ holds, the unawareness price coincides with the discrimination-free price. However, this condition often fails in practice due to omitted variables. In such cases, it is essential to reorganize the trade-off between the two types of fairness embedded in the discrimination-free premium  $h^*(X)$, equity-based and risk-based fairness. The former aims to prevent differentiation in pricing based on the sensitive attribute D, while the latter concerns the across-group subsidization that potentially arises when the sensitive attribute $D$ is excluded from pricing. The overall value of the discrimination-free premium ultimately depends on the extent of cross-subsidization. In practice, insurers address this issue by gathering additional data from various sources to account for potential omitted variables in their pricing models.”

---

> > > > ### Comment · Reviewer_2RA7 · 2024-11-26
> > > >
> > > > I have updated my score. Thanks for your patient explanations. I'm still skeptical of privacy math but I'm now confident this paper is more than just an exercise in it!

---

> > > > > ### Author Response · Authors · 2024-11-26
> > > > > **Thank You Note to Reviewer 2RA7**
> > > > >
> > > > > Dear Reviewer 2RA7,
> > > > >
> > > > > Thank you for all your prompt responses that kept the discussion running so that we had the opportunity to clarify things!
> > > > >
> > > > > We again express our gratitude for all your insightful comments, suggestions, and acknowledgment of our responses.

---

### Official Review · Reviewer_nCT9 · 2024-11-03

**Soundness:** 4
**Presentation:** 3
**Contribution:** 3
**Rating:** 8
**Confidence:** 3

**Summary:**

This paper addresses fairness in insurance pricing where the use of sensitive attributes like race and gender is restricted. Recent regulatory demands requires approaches that prevent discrimination without directly using sensitive data. The authors propose a multi-party framework where insurers collaborate with a trusted third party (TTP) and the sensitive attributes influence pricing only indirectly.

The main contributions of the paper are: (i) The authors introduce a framework that enables insurers to calculate fair premiums without direct access to sensitive attributes, while the TTP uses a noised version of sensitive features to generate "discrimination-free" insurance prices. (ii) The authors provide theoretical guarantees for two settings -- when the noise of the sensitive features is known and unknown to the TTP. (iii) The paper demonstrates the method’s effectiveness in two insurance datasets, showing its robustness to noise and the impact of noise estimation errors on performance.

**Strengths:**

To start, I think this paper is a very strong paper. It studies an important research question -- fairness in insurance pricing, which is a domain that typically emphasizes actuarial fairness over algorithmic fairness. The authors introduce an innovative framework that adapts to regulatory requirements by using privatized, noised sensitive attributes for TTP to calculate discrimination-free premiums. This application of differential privacy within a multi-party setup in the insurance sector is novel and distinguishes this work from other fairness research.

What I like about the paper the most is its high quality in its technical rigor and thoroughness. The authors offer theoretical guarantees for scenarios with known and unknown noise rates, with clear delineation of assumptions and the rigorous derivation of theoretical results.

The paper is also well-organized and easy to follow. Key concepts, such as "discrimination-free pricing," are defined early on, and the technical sections are structured nicely, which allows readers to follow the framework’s development from problem formulation to implementation. The explanations of complex privacy mechanisms, including local differential privacy and its implications for sensitive data handling, are accessible even for readers less familiar with privacy-preserving machine learning.

The paper also provides an in-depth empirical analysis section that effectively demonstrates the method’s applicability and limitations, contributing to the clarity of the results.

**Weaknesses:**

1) The paper's setting relies on the feasibility of the framework via data sharing with TTP. What if this framework is not feasible, e.g., using a TTP is not feasible? Would the authors be able to comment on something along this line and also comment on the flexibility of their proposed framework under different regulatory requirements?

2) The tightness of the upper bound provided in the two theorems.

3) It would be nice to add comparisons to existing fairness methods or baseline models for insurance pricing.

4) The study focuses on the US Health Insurance and Auto Insurance datasets, but additional datasets -- particularly those representing other insurance types or more complex demographic distributions -- could add value.

5) The paper could benefit from a more explicit discussion of its limitations and future directions.

**Questions:**

1) Could the authors provide more insights into the tightness of the upper bound presented in Theorems 4.3 and 4.5? For example, how does the gap between the empirical risk R^LDP (f)  and  R(f^*)  behave as noise decreases?

2) Do the authors see potential for this framework to be applied in domains beyond insurance? For instance, could it be adapted for fields with similar regulatory constraints on fairness and sensitive data, such as finance or healthcare?

3) The framework currently focuses on discrete sensitive attributes. Could the authors elaborate on how it might extend to continuous sensitive attributes, or discuss challenges that may arise?

4) Could the authors explicitly discuss known limitations of the proposed approach?

---

> ### Author Response · Authors · 2024-11-20
> **Response to Reviewer nCT9 (Part 1)**
>
> Dear Reviewer nCT9:
>
> Thank you for taking the time to review our paper and for providing thoughtful and constructive feedback. We have carefully considered your comments and incorporated them into our revisions.
>
> Below, you will find our detailed responses to each of your questions and concerns.
>
> $\textbf{Q1:}$ "The paper's setting relies on the feasibility of the framework via data sharing with TTP. What if this framework is not feasible, e.g., using a TTP is not feasible? Would the authors be able to comment on something along this line and also comment on the flexibility of their proposed framework under different regulatory requirements?"
>
> $\textbf{A1:}$ Thank you for your thoughtful question. We address your concerns in three parts:
>
> First, the protocol between insurers and TTPs is already well-established in the insurance market. The proposed method can seamlessly integrate with these existing protocols, making it readily applicable. For example, Insurers, particularly small to mid-sized, commonly purchase industry-wide data, process it using credibility techniques, and then transfer the processed data to a data service platform (e.g., DataRobot) for pricing. The proposed method fits naturally into this workflow by allowing insurers to collect both sensitive and non-sensitive attributes and forward the data to a third-party vendor for pricing algorithm execution. As another example, when entering new markets where insurers lack historical data, they often rely on external parties, such as consulting firms or data vendors. The proposed method is applicable in this scenario if the insurer can only gather non-sensitive attribute information for potential customers in the new market. The insurer would send this transformed information, $T(X)$, to an external vendor to perform the pricing process using the proposed procedure.
>
> Second, for large insurers, such as State Farm or Allstate, that possess significant resources, including research teams and computational infrastructure, the framework remains feasible even without a TTP. In this setting, the insurer can independently obtain sensitive attributes or derive them through auxiliary means and implement the proposed method in-house. This in-house implementation represents a special case of the proposed framework, demonstrating its adaptability to scenarios where a TTP is not involved.
>
> Third, the proposed framework is highly adaptable to varying regulatory environments. Regulatory tolerance for model transparency can differ across insurance products or geographical regions. In our proposed framework, the model transparency is governed by the transformation $T$ and the hypothesis class $\mathcal{F}$. By allowing $T$ and $\mathcal{F}$ to be arbitrarily defined, the framework can adapt to diverse regulatory constraints, whether they demand greater transparency or prioritize other considerations. This versatility makes it practical and robust across a wide range of regulatory requirements.
>
> As a summary, the above examples illustrate the flexibility and practicality of our framework under different scenarios, ensuring its feasibility and applicability regardless of whether a TTP is involved or specific regulatory constraints are in place.
>
> $\textbf{Q2:}$ "The tightness of the upper bound provided in the two theorems."
>
> $\textbf{A2:}$ Thank you for your comment. We note that the upper bound can be loose when $C_1$ is large and $C_1$ is fully governed by the noise rate. We also note that the upper bound can be loose when $|D|$ is large. However, in many insurance applications, $|D|$ is relatively small when the sensitive attributes are demographic characteristics such as gender or ethnicity. For sensitive attributes with a large number of levels, techniques such as categorical embedding are often employed to reduce the cardinality of the cardinality of $|D|$. This is because we often observe insufficient data on some levels and we expect a sparse structure in terms of the relationship between diﬀerent levels (i.e. there are subgroups among the large number of levels)[1]. Therefore, a dimension reduction via the categorical embedding technique can make the upper bound more informative when $|D|$ is large (please see remark, line 311-312, on page 6 of the revised paper).
>
>
> [1] Shi, P., & Shi, K. (2023). Non-Life Insurance Risk Classification Using Categorical Embedding. North American Actuarial Journal, 27(3), 579–601.

---

> > ### Comment · Reviewer_nCT9 · 2024-11-27
> >
> > Thank you for the detailed response. This is a really good paper. As I already recommended accept, I will keep my rating.

---

> > > ### Author Response · Authors · 2024-11-27
> > > **Thank You Note to Reviewer nCT9**
> > >
> > > Dear Reviewer nCT9,
> > >
> > > We again express our gratitude for all your insightful comments, suggestions, and acknowledgment of our responses.

---

> ### Author Response · Authors · 2024-11-20
> **Response to Reviewer nCT9 (Part 2)**
>
> $\textbf{Q3:}$ "It would be nice to add comparisons to existing fairness methods or baseline models for insurance pricing."
>
> $\textbf{A3:}$ Thank you for your insightful comments. First, we would like to highlight that fairness in insurance is a relatively new topic, and notions of fairness specific to insurance are rarely addressed in the existing literature. As a result, there is not yet a well-established line of research proposing methods to achieve fairness in insurance in the way we have framed it. However, the insurance sector plays a critical role in the modern economy, contributing approximately 3-4% of GDP in the U.S., which underscores the importance of our work. In our empirical analysis, we used the case of true sensitive attributes as a baseline for comparison to provide a meaningful benchmark.
>
> Second, to clarify the distinction between the fairness notion proposed in discrimination-free price (Definition 3.3), and other fairness notions in the newly formed fair insurance pricing literature. The key challenge lies in accurately estimating $\mu(X, D)$. There is, however, some flexibility in determining the final discrimination-free price $h^*(X)$, depending on the choice of $\mathbb{P}^*(d)$, as discussed in the beginning of Section 4 (line 189-193, on page 4 highlighted in red). In contrast, other fairness notions in insurance do not allow for such flexibility in their final fair price—or at least do not define it in the same manner. This fundamental difference makes direct comparisons less informative. Therefore, instead of making potentially uninformative comparisons, we chose to focus on further investigations of our proposed method, such as analyzing the impact of noise rate estimation errors and addressing imbalanced distributions. These efforts provide deeper insights into the robustness and practical application of our approach.
>
> $\textbf{Q4:}$ "The study focuses on the US Health Insurance and Auto Insurance datasets, but additional datasets -- particularly those representing other insurance types or more complex demographic distributions -- could add value."
>
> $\textbf{A4:}$ We have focused on the auto insurance and health insurance datasets in our empirical analysis for two key reasons: First, both sectors are highly regulated compared to other lines of insurance, such as commercial insurance, making them ideal settings for studying fairness in insurance.
>
> Second, both auto and health insurance are significant in terms of market size. Health insurance, for instance, is one of the largest sectors in the U.S. insurance market, with a substantial impact on the economy and the well-being of the population. As of 2023, over 91% of Americans had health insurance coverage, and the health insurance market was valued at approximately $1.1 trillion in premiums. Similarly, the auto insurance market is the largest segment in property and casualty (P&C) insurance in the U.S., accounting for 35-40% of the total market in terms of premium volume. Given their scale and regulatory importance, these sectors provide a representative basis for our study.
>
> We do agree and fully understand that additional analysis using data from other types of insurance would provide valuable insights. However, due to time constraints, we leave this exploration as part of future investigations.
>
> $\textbf{Q5:}$ "The paper could benefit from a more explicit discussion of its limitations and future directions."
>
> $\textbf{A5:}$ In the revised paper, we have included a more explicit discussion of the limitations of the proposed method and outlined potential future research directions, which states "The main limitation of our work is that the risk bound in Theorem 4.3 and Theorem 4.5 may be less informative in certain scenarios. For instance, regulatory transparency requirements might prevent insurers from applying dimension-reduction techniques to high-cardinality sensitive attributes. Future work could extend the framework to accommodate continuous sensitive attributes and adapt it to other fields with similar regulatory constraints." (this is included in the revision line 536-539 on page 10, highlighted in red).
>
> Despite the aforementioned limitations, we emphasize that this study is among the first to address fairness notions in insurance, a critical sector of the modern economy. We anticipate this work will serve as a foundation and catalyst for further investigations in this important domain.

---

> ### Author Response · Authors · 2024-11-20
> **Response to Reviewer nCT9 (Part 3)**
>
> $\textbf{Q6:}$ "Could the authors provide more insights into the tightness of the upper bound presented in Theorems 4.3 and 4.5? For example, how does the gap between the empirical risk R^LDP (f) and R(f^*) behave as noise decreases?"
>
> $\textbf{A6:}$ in Theorem 4.5, as the noise in estimating $\pi$ decreases, the relaxation term $\tilde{\epsilon}$ correspondingly diminishes, assuming the number of groups, $n_1$, remains fixed. Generally, reduced noise facilitates achieving a tighter bound, which will asymptotically approach the original bound described in Theorem 4.3.
>
> $\textbf{Q7:}$ "Do the authors see potential for this framework to be applied in domains beyond insurance? For instance, could it be adapted for fields with similar regulatory constraints on fairness and sensitive data, such as finance or healthcare?"
>
> $\textbf{A7:}$ Thank you for your insightful question. We hadn't fully considered this application until your comment, but we do see the potential for extending this framework to domains beyond insurance. Specifically, fields like finance and healthcare share similar regulatory constraints around fairness and the use of sensitive data.
> For instance, in finance, credit scoring models often rely on sensitive data such as income, employment status, and even demographic information. Regulators, such as the Consumer Financial Protection Bureau (CFPB), impose strict rules regarding fairness to ensure that these models do not unfairly disadvantage certain groups. Similarly, in healthcare, the use of sensitive health data, such as patient demographics and medical history, is highly regulated under laws like HIPAA, and any predictive models in healthcare need to comply with fairness regulations.
> In these contexts, our framework could potentially be adapted to ensure fairness in decision-making processes while protecting sensitive data. However, we recognize that the notions of fairness in other industries may differ from those in the insurance sector, which could present challenges when directly applying our proposed method. This, in turn, makes it an interesting avenue for future research.
>
> $\textbf{Q8:}$ "The framework currently focuses on discrete sensitive attributes. Could the authors elaborate on how it might extend to continuous sensitive attributes, or discuss challenges that may arise?"
>
> $\textbf{A8:}$ Thank you for your thoughtful comment. From a technical perspective, we believe it is indeed possible to extend the framework to account for continuous sensitive attributes. Specifically, a potential extension is as follows: in Eq. (1), we reformulate the score function $f_k(X)$ as a continuous version, denoted by $f(X,t)$. Given that $D$ is continuous, we must replace the transition probability matrix with a transition probability kernel. With this setup, we then aim to optimize $f(X,\cdot)$ as a function, where, in Eq. (2), the summation over the subscripts $k$ is replaced by integration.
>
> However, we immediately see challenges regarding:
> 1) selection of the class of continuous distributions for $D$.
> 2) the optimization of $f(X,t)$ within a continuous function class.
> 3) the need for a well-defined noise mechanism that properly establishes a transition probability kernel.
>
> We acknowledge that this would be an interesting and valuable extension of our current work. However, we also recognize that it is a non-trivial task, requiring different techniques. Specifically, it may involve substantial technical details and theories from functional analysis and the calculus of variations.
>
> From a practical standpoint, however, the discrete framework aligns more closely with current pricing practices in the insurance industry. Specifically, insurers typically use discrete variables to define rating classes. In cases where continuous variables are involved, it is common practice to discretize these variables, converting them into categorical ones for ease of use in pricing models.
>
> $\textbf{Q9:}$ "Could the authors explicitly discuss known limitations of the proposed approach?"
>
> $\textbf{A9:}$ In the revised paper, we have included a more explicit discussion of the limitations of the proposed method and outlined potential future research directions, which states "The main limitation of our work is that the risk bound in Theorem 4.3 and Theorem 4.5 may be less informative in certain scenarios. For instance, regulatory transparency requirements might prevent insurers from applying dimension-reduction techniques to high-cardinality sensitive attributes. Future work could extend the framework to accommodate continuous sensitive attributes and adapt it to other fields with similar regulatory constraints." (this is included in the revision line 536-539 on page 10, highlighted in red).

---

### Official Review · Reviewer_oqij · 2024-11-04

**Soundness:** 3
**Presentation:** 3
**Contribution:** 2
**Rating:** 8
**Confidence:** 3

**Summary:**

This work studies the problem of providing discrimination-free insurance prices when information about true sensitive attributes is hidden from the insurer---in particular, it is held by a trusted third-party which, for example, may add noise to ensure that sensitive attributes are differentially-private. In addition to showing risk bounds for both known and unknown noise rates, the authors conduct thorough experimental evaluations for how properties of the problem instance and hyperparameters used in the algorithm affect overall performance.

**Strengths:**

* Exposition in section 4 is straightforward (especially leading up to Theorem 4.3).
* Experimental evaluation is thorough and well motivated.

**Weaknesses:**

* I'm not sure how realistic/practical this problem setup is at a high level; do such protocols currently exist?
* Risk bounds in section 4 seem to follow pretty 'classic' concentration-type arguments, esp. in the dependence on VC(F). This is fine, though it does mean the interpretation should be more qualitative (scaling wrt $n$ and noise level) than quantitative

Presentation comments:
* it could be helpful to have a diagram illustrating the protocol for interaction between insurer and TTP.
* Maybe worth noting that because risk is defined wrt an arbitrary (possible per-group, if I understand correctly?) $L$, minimizing risk of $f$ is sufficient to optimize for (e.g.) the ideal price $h^*$ or whatever downstream utility the insurer gets.

**Questions:**

* Why is the transformation $T(X)$ necessary?

---

> ### Author Response · Authors · 2024-11-20
> **Response to Reviewer oqij (Part 1)**
>
> Dear Reviewer oqij:
>
> Thank you for taking the time to review our paper and for providing thoughtful and constructive feedback. We have carefully considered your comments and incorporated them into our revisions.
>
> Below, you will find our detailed responses to each of your questions and concerns.
>
> $\textbf{Q1:}$ "I'm not sure how realistic/practical this problem setup is at a high level; do such protocols currently exist?"
>
> $\textbf{A1:}$ This is a great question. The answer to your question is yes. The protocol between insurers and TTPs is already well-established in the insurance market. The proposed method can seamlessly integrate with these existing protocols, making it readily applicable. Below, we describe three scenarios where the proposed method can be implemented using current frameworks:
> 1) $\textbf{Large-Size (Tier-1) Insurers:}$ Major insurance companies (e.g., State Farm, Allstate) typically have the resources, such as dedicated research teams and computational infrastructure, to handle pricing in-house. However, they often supplement their proprietary data with third-party data. In this scenario, insurers can leverage existing protocols to obtain sensitive attributes from third parties. The proposed method is directly applicable here as a specific case of the broader framework.
> 2) $\textbf{Small to Mid-Sized Insurers:}$ For smaller insurers, maintaining an in-house pricing team is often not cost-effective. These companies commonly purchase industry-wide data, process it using credibility techniques, and then transfer the processed data to a data service platform (e.g., DataRobot) for pricing. In this context, the proposed method can be applied by having the insurer collect both sensitive and non-sensitive attributes and forward this data to a third-party vendor for pricing algorithm execution.
> 3) $\textbf{New Market Development:}$ When entering new markets where insurers lack historical data, they often rely on external parties, such as consulting firms or data vendors. The proposed method is applicable in this scenario if the insurer can only gather non-sensitive attribute information for potential customers in the new market. The insurer would send this transformed information, $T(X)$, to an external vendor to perform the pricing process using the MTPT procedure.
>
> These examples demonstrate the versatility and practicality of the proposed method within the existing insurer-TTP interaction protocols.
>
> $\textbf{Q2:}$ "Risk bounds in section 4 seem to follow pretty 'classic' concentration-type arguments, esp. in the dependence on VC(F). This is fine, though it does mean the interpretation should be more qualitative (scaling wrt $n$ and noise level) than quantitative"
>
> $\textbf{A2:}$ We agree that the main results rely on classic concentration-type arguments, particularly in their dependence on VC(F). As you have pointed out, the nature of these arguments naturally suggests that the interpretation should be more qualitative rather than strictly quantitative. We have revised the paper accordingly to emphasize this perspective (please see the revision in the remarks line 359-369 on page 7 highlighted in red).
>
> Although the tools used in our work are ‘classic,’ we emphasize that this study is among the first to address fairness notions in insurance, a critical sector of the modern economy. We anticipate this work will serve as a foundation and catalyst for further investigations in this important domain.
>
> $\textbf{Q3:}$ "It could be helpful to have a diagram illustrating the protocol for interaction between insurer and TTP."
>
> $\textbf{A3:}$ Thank you so much for the excellent suggestion. In the revised paper, we have included a diagram illustrating the protocol for interaction between the insurer and the TTP (see Figure 1 on page 2 in the revised paper).
>
> $\textbf{Q4:}$ "Maybe worth noting that because risk is defined wrt an arbitrary (possible per-group, if I understand correctly?) $L$, minimizing risk of $f$ is sufficient to optimize for (e.g.) the ideal price $h^*$ or whatever downstream utility the insurer gets."
>
> $\textbf{A4:}$ Thank you so much again for the excellent suggestion. Yes, your understanding is correct. In the revised paper, we have included a comment, which states: “The optimized score functions $f_k$’s are independent of the specific form of $h^*$. More generally, the optimized $f_k$’s can be directly applied to any downstream tasks that do not depend on the optimization procedure of $f_k$’s.” See Remarks 3, line 223-224 on page 5.

---

> ### Author Response · Authors · 2024-11-20
> **Response to Reviewer oqij (Part 2)**
>
> $\textbf{Q5:}$ "Why is the transformation $T(X)$ necessary?"
>
> $\textbf{A5:}$ The use of transformation $T$ is driven by practical, real-world considerations. For instance, insurers may seek to protect their pricing strategies when sharing information with a trusted third party (TTP). In such cases, a simple transformation can effectively obscure the original rating variables. Conversely, an insurer might employ a more complex transformation to enhance risk assessment. This transformation could be learned using auxiliary data that is larger and more comprehensive than the data shared with the TTP for a specific pricing task.
>
> Furthermore, we presented the method within a general framework. In cases where the transformation is unnecessary for specific applications, the identity transformation can be employed for $T$.

---

> ### Author Response · Authors · 2024-11-25
> **Follow-up on Response to Reviewer oqij**
>
> Dear Review oqij:
>
> Thank you once again for your valuable feedback and insightful comments on our paper. We hope you have had the opportunity to review our revised manuscript and our detailed responses to each of your remarks.
>
> We would like to kindly remind you that the discussion period concludes on $\textbf{November 26th}$. If you have any additional questions or concerns regarding our revisions, please let us know at your earliest convenience so we can address them promptly.
>
> If you find our explanations and revisions satisfactory, we kindly ask you to consider revising your overall rating for our paper.

---

> ### Comment · Reviewer_oqij · 2024-11-25
>
> Hi and apologies for the delay in responding. These answers make sense to me and I appreciate the revisions - I will be revising my score upwards. Figure 1 looks great! I think it would be also helpful to include your answer to Q1 somewhere in the related work or problem formulation section just to emphasize that your work is building on mechanisms that currently exist.

---

> > ### Author Response · Authors · 2024-11-26
> > **Thank You Note to Reviewer oqij**
> >
> > Dear Reviewer oqij,
> >
> > Thank you again for your valuable comments. In the revised paper, we have added a discussion to emphasize that the proposed work builds on the well-established insurer-TTP protocols already in place in the insurance market (please see line 85-95 on page 2, highlighted in red).
> >
> > We again express our gratitude for all your insightful comments, suggestions, and acknowledgment of our responses.

---

### Official Review · Reviewer_m1qD · 2024-11-07

**Soundness:** 4
**Presentation:** 3
**Contribution:** 3
**Rating:** 6
**Confidence:** 2

**Summary:**

This paper considers fairness insurance pricing problems. The method proposes to achieve actuarial fairness in insurance pricing. Actuarial fairness is different from conventional machine learning fairness concepts, like demographic parity and equalized odds. In insurance pricing, existing works mainly consider three methodologies in solving fairness: counterfactual approach, group fairness approach, and probabilistic approach. However, all aforementioned methods requiring direct access of sensitive information might not be available due to regulation.  This paper proposes a method that introduces the trusted third party (TTP) that deals with noiseless or noisy sensitive information, allowing discrimination-free premium without direct access to sensitive information.

**Strengths:**

This paper addresses a significant real-world challenge: achieving discrimination-free insurance pricing. The proposed method is not only grounded in solid mathematical foundations but also applicable to practical applications. It has been thoroughly evaluated through comprehensive experiments on scenarios with both known and unknown noise rates.

**Weaknesses:**

Please see the questions.

**Questions:**

1. question regarding the problem setting: Does the insurer have access to sensitive attributes, but only in an indirect manner, such that they cannot directly access these sensitive attributes? If so, the TTP may need to generate a fair premium without directly knowing the sensitive attributes. In that case, why do we still need the TTP? Could the insurer implement the method directly to generate the premium on their side instead?

2.  Why does integral over $d$ with measure $\mathbb{P}^*(d)$ bring discrimination-free price if $X$ contains information of $d$.

3. Could the author elaborate on Theorem 4.5? What does $\tilde{\epsilon}$ represent here? When compareTheorem 4.5 to Theorem 4.3, the only difference is this term. Does it represent the error from estimating the error rate $\pi$ and $\bar{pi}$?

4. Does this condition $M_g+\frac{C_1+\theta}{ln2}>\tilde{\epsilon}>\theta$ have some meaning or is it simply a technical assumption?
.

---

> ### Author Response · Authors · 2024-11-20
> **Response to Reviewer m1qD**
>
> Dear Reviewer m1qD:
>
> Thank you for taking the time to review our paper and for providing thoughtful and constructive feedback. We have carefully considered your comments and incorporated them into our revisions.
>
> Below, you will find our detailed responses to each of your questions and concerns.
>
> $\textbf{Q1:}$ "question regarding the problem setting: Does the insurer have access to sensitive attributes, but only in an indirect manner, such that they cannot directly access these sensitive attributes? If so, the TTP may need to generate a fair premium without directly knowing the sensitive attributes. In that case, why do we still need the TTP? Could the insurer implement the method directly to generate the premium on their side instead?"
>
> $\textbf{A1:}$ You are correct that the insurer could implement the proposed method directly without the TTP, provided that they have access to the sensitive attributes. As we clarified in the paper (line 92-96 on page 2 highlighted in red), we consider this scenario as a nested case of our multi-party framework.
>
> $\textbf{Q2:}$ "Why does integral over $d$ with measure $\mathbb{P}^*(d)$ bring discrimination-free price if $X$ contains information of $d$."
>
> $\textbf{A2:}$ Our work builds on the concept of the discrimination-free premium introduced by Lindholm et al (2021), which requires that the premium must be independent of sensitive attributes, both directly and indirectly. Since the measure $\mathbb{P}^*(d)$ is independent of $X$, the resulting premium (Definition 3.3) satisfies this requirement and is therefore discrimination-free. Conceptually, the discrimination-free price (Definition 3.3) can be viewed as a “change of measure” version of the unawareness premium (Definition 3.2), where $\mathbb{P}^*(d)$ eliminates the indirect dependence of $\mathbb{P}(d|X)$ on $X$.
>
> $\textbf{Q3:}$ "Could the author elaborate on Theorem 4.5? What does $\tilde{\epsilon}$ represent here? When comparing Theorem 4.5 to Theorem 4.3, the only difference is this term. Does it represent the error from estimating the error rate $\pi$ and $\bar{\pi}$?"
>
> $\textbf{A3:}$ Yes, your observation is correct. In Theorem 4.3, the only term that depends on  $\pi$ is $C_1$. Since $\pi$ is known under this setting, $C_1$ is consequently a known constant. However, In Theorem 4.5, $\pi$ is unknown and is replaced by an estimate $\hat{\pi}$,  we must account for the estimation error. Given $C_1$ is a non-linear function of $\pi$, it is necessary to bound the entire term (i.e. the gap between $C_1$ and $\hat{C_1}$), rather than just error in $\pi$.
>
> To accomplish this, we establish a bound on the gap between $C_1$ and $\hat{C}_1$ by $\tilde{\epsilon}$ with probability $1 - \delta$, utilizing the sub-exponential assumption along with Bernstein's inequality. This randomness is then represented by $\tilde{\epsilon}$ in Theorem 4.5. In the revised paper, we included some comments (see Remark 1 line 359-360 on page 7, highlighted in red) to emphasize this point.
>
> As a side note, the term $\tilde{\epsilon}$ can be sufficiently reduced by making an appropriate choice on $n_1$, as demonstrated in Section 5 in the revised paper.
>
> $\textbf{Q4:}$ "Does this condition $M_g+\frac{C_1+\theta}{\ln 2}>\bar{\epsilon}>\theta$ have some meaning or is it simply a technical assumption?"
>
> $\textbf{A4:}$ This condition is specifically employed to circumvent the discussion about which of the two terms in the Bernstein inequality is the minimal one (see the first part of the proof on the bottom of Page 24 in the Appendix). Adopting this approach allows for a cleaner and more concise presentation of the result without getting entangled in complex considerations. Though it might lead to a slightly looser bound, it is a common practice when working with the Bernstein inequality.

---

> ### Author Response · Authors · 2024-11-25
> **Follow-up on Response to Reviewer m1qD**
>
> Dear Review m1qD:
>
> Thank you once again for your valuable feedback and insightful comments on our paper. We hope you have had the opportunity to review our revised manuscript and our detailed responses to each of your remarks.
>
> We would like to kindly remind you that the discussion period concludes on $\textbf{November 26th}$. If you have any additional questions or concerns regarding our revisions, please let us know at your earliest convenience so we can address them promptly.
>
> If you find our explanations and revisions satisfactory, we kindly ask you to consider revising your overall rating for our paper.

---

> ### Author Response · Authors · 2024-11-27
> **Follow-up on Response to Reviewer m1qD**
>
> Dear Review m1qD:
>
> Thank you once again for your valuable feedback and insightful comments on our paper. We hope you have had the opportunity to review our revised manuscript and our detailed responses to each of your remarks.
>
> We would like to kindly remind you that the last day authors can upload a revised PDF of the paper is $\textbf{November 27th}$. If you have any additional questions, concerns, or suggestions regarding our revisions, please let us know at your earliest convenience so we can address them promptly.
>
> If you find our explanations and revisions satisfactory, we kindly ask you to consider revising your overall rating for our paper.

---

> ### Author Response · Authors · 2024-12-02
> **Follow-up on Response to Reviewer m1qD**
>
> Dear Review m1qD:
>
> Thank you once again for your valuable feedback and insightful comments on our paper. We hope you have had the opportunity to review our revised manuscript and our detailed responses to each of your remarks.
>
> We would like to kindly remind you that the last day the reviewer may post a message to authors is $\textbf{December 2nd (AOE)}$. If you have any additional questions or concerns regarding our revisions, please let us know at your earliest convenience so we can address them promptly.
>
> If you find our explanations and revisions satisfactory, we kindly ask you to consider revising your overall rating for our paper.

---

### Author Response · Authors · 2024-11-20
**Gratitude to All Reviewers and Summary of Updates**

We would like to express our gratitude to all reviewers for their insightful feedback and valuable suggestions, which have significantly improved the quality and presentation of our paper.

We have thoroughly reviewed all comments and suggestions and incorporated them into the revised version of our paper. Below, we outline the key updates made:

1) We have included a diagram to illustrate the interaction between the insurer and TTP in our multi-party training framework, aiming to enhance understanding.
2) We have emphasized both the theoretical innovation and the practical applicability of the proposed framework, highlighting its ease of use in real-world settings. Additionally, we have added several remarks to clarify and interpret our results.
3) We have included a discussion of the limitations of our method and provided insights into future research directions.

We hope the revision clarifies the goals and contributions of our research, and that you now find it a suitable contribution to ICLR 2025.

---

### Meta-Review · Area_Chair_vZXk · 2024-12-21

**Metareview:**

This paper presents a method to learn models to price insurance policies in a way that does not discriminate between subgroups. The authors consider a setting where an insurance provider must learn their model using a dataset of noisy privatized sensitive attributes. They propose algorithms for settings  with known and unknown noise rates, and evaluate their models on 2 datasets.

**Decision**: This paper initially received mixed feedback from reviewers due to concerns about its motivation, formulation, and potential impact. Over the rebuttal and discussion period, the paper ended up with uniformly positive scores from all reviewers. However, all reviewers had low confidence scores. Given the low confidence scores, I reviewed the submission in detail -- reading the paper, the reviews, and the authors' responses.

Given my reading, I am unfortunately recommending rejection. My decision is based on major deficiencies in evaluation and exposition that affect significance, clarity, and (potentially) soundness of the paper. I describe these issues in detail below. At a high level, the issues in evaluation were missed by reviewers, while the issues in exposition were discounted under the assumption that they could be addressed through revisions. While I agree that some of the issues could be addressed through revisions, the revisions should be checked by another round of peer review. This is, in part, because the integrity of the contributions now depends on the experiments (whose outcome is uncertain), and because the changes in exposition depend on the willingness and judgement of authors. Accepting the paper as-is runs the risk that the method may not be as reliable as claimed, and that it will have limited impact because it will remain difficult to parse.

**Evaluation**

1. The experiments only evaluate their method on 2 datasets. The current set of results only shows that the test loss converges in settings with more than 2 groups.

2. The experiments do not provide a clear picture of how the method will affect premiums between groups. This is a major omission given the goal of the paper.

3. The paper does not discuss or study potential limitations. Specifically, we are not sure of what can/cannot go wrong, how bad it may be, and whether we should attribute it to noise or other decisions in algorithm development. What is missing is an experiment that addresses questions raised by 2RA7 on synthetic data where the authors know ground truth and that studies the effect on premiums.

4. The experiments report all results in ways that compromise clarity (see 2RA7's comment about "convergence"). As it stands, all methods are shown as convergence plots. This makes it hard to evaluate the measures that we care about - i.e., the performance of the final model -- and overemphasizes convergence (c.f., the changes in distribution between groups).

5. The discussion in this section is lacking. Many claims focus on superficial behavior and are not always clearly linked to the plots that we see.

Given this, I would recommend: (1) evaluate the model on more datasets (ideally 4-5, and one with multiple groups); (2) report results using a table of summary statistics for the final model from each method on each dataset. This should include more multiple metrics, and allow readers to make comparisons across methods and metrics; (3) support each claim and takeaway to specific results and measures in this table; (4) move all results to the supplement.

**Exposition**

6. The paper is written in a way that makes it difficult for researchers with expertise to understand what is being done, why it is required, and what impact it would have. As it stands, the abstract, introduction, and problem statement are missing information to motivate the problem and the proposed solution (see e.g., questions from 2RA7 and oqij). The experiments are missing information to understand the impact of the method and its limitations (e.g., on the distribution of premiums). These issues are not about writing but rather technical communication. In this case, both the submission and revisions point to a lack of editorial judgement that underscores the need of a second round of review. For example, the core contribution -- i.e., the algorithms -- are tucked away in the supplement and replaced by information that is cursory. As another example, the edits made to address misleading statements about generalization bounds in the experiments consist of vague sentences in the conclusion (note the revision still contains the same statements).

**Additional Comments On Reviewer Discussion:**

My recommendation is based - in part - on the importance and viability of addressing these issues. I see these issues as critical barriers for acceptance for two reasons. First, because it is hard for any reviewer to evaluate significance and soundness when a manuscript is missing so much of this information. Second, because it will ultimately determine the impact and reception of the paper. As stated by 2RA7 -- without these changes -- the paper remains an "exercise in privacy math."

Over the rebuttal period, the authors provided clear and concise answers to questions from 2RA7 and oqij. The responses left reviewers (and myself) with the impression that the authors would address critical issues related to exposition and clarity - see e.g. Reviewer 2RA7 who emphasizes that their score is contingent on integrating some of these points into the final paper.

What I am concerned about are statements from the authors that they have addressed these issues in their revised manuscript and that the quality of their work has "significantly improved." As it stands, the revised manuscript -- which I read in preparing the meta-review -- still suffers from many of the problems described above, and the statements from the authors suggest that the revisions should be validated by another round of peer review.

---

### Decision · Program_Chairs · 2025-01-22

Reject